# Altered functional interactions between CFTR disease mutants ΔF508 and G551D and the protein kinase A catalytic subunit

Olivér Závoti[1,2], Márton A. Simon[1,2] and László Csanády[1,2] (iD)

[1]*Department of Biochemistry, Semmelweis University, Budapest, Hungary*
[2]*HUN-REN-SE Ion Channel Research Group, Semmelweis University, Budapest, Hungary*

Handling Editors: Peying Fong & Péter Hegyi

The peer review history is available in the Supporting Information section of this article (https://doi.org/10.1113/JP290798#support-information-section).

**Abstract figure legend** Pathogenic mutations ΔF508 and G551D of the cystic fibrosis transmembrane conductance regulator (CFTR) channel alter its interactions with protein kinase A (PKA). Top left, for both mutants, but not for wild-type (WT) CFTR, non-catalytic stimulation of channel activity by PKA is larger in the presence of $N^6$-(2-phenylethyl)-ATP (P-ATP; dark grey bars) than in the presence of ATP (light grey bars). Bottom, compared with WT CFTR (blue current traces), for both mutants (black current traces), the time course of activation by PKA is slowed, and the current fraction surviving PKA removal is reduced. Top right, for phosphorylated ΔF508 CFTR, non-catalytic activation by PKA is preserved in the presence of the potentiator ivacaftor (left) or elexacaftor (centre), but undetectable when both drugs are applied simultaneously (right).

O. Závoti and M. A. Simon contributed equally to this work.

**Abstract** The epithelial anion channel cystic fibrosis transmembrane conductance regulator (CFTR) is activated by cAMP-dependent protein kinase (PKA). PKA stimulates CFTR channels through two mechanisms: non-catalytically, by binding to the channel, and catalytically, by phosphorylating its regulatory (R) domain. CFTR mutations that reduce channel activity cause cystic fibrosis (CF), but clinically used modulator drugs that boost channel function can alleviate disease symptoms. The two common CF mutations, ΔF508 and G551D, have been reported to impair CFTR channel activation by PKA, but the mechanisms remain unclear. Here, we aimed to understand how the mutations impact non-catalytic *vs.* catalytic channel activation by PKA and how these two processes are modulated by clinically used potentiator drugs. Using current recordings from excised inside-out membrane patches superfused with the purified catalytic subunit of PKA, we confirm slowed PKA-dependent activation for both mutants but demonstrate intact binding affinity for the kinase. Furthermore, we find that non-catalytic activation dominates overall channel activity for both mutants and can be strongly enhanced by stabilization of the NBD1–NBD2–TMD interface using the ATP analogue $N^6$-(2-phenylethyl)-ATP. For both mutants, the clinically used potentiator drug combination elexacaftor + ivacaftor boosts catalytic channel activation by PKA more efficiently than non-catalytic activation. For ΔF508 CFTR, elexacaftor + ivacaftor evokes substantial PKA-independent channel activity and entirely suppresses non-catalytic activation by PKA. These findings help us to understand the activation defects caused by two common CF mutations and suggest room for further improvement of potentiator drugs currently used in CF therapy.

(Received 22 December 2025; accepted after revision 25 February 2026; first published online 1 April 2026)

**Corresponding author** L. Csanády: Department of Biochemistry, Semmelweis University, Tűzoltó u. 37-47, 1094 Budapest, Hungary. Email: csanady.laszlo@semmelweis.hu

## Key points

- Protein kinase A (PKA) activates the epithelial anion channel cystic fibrosis transmembrane conductance regulator (CFTR) through two mechanisms: non-catalytically, by binding to the channel, and catalytically, by phosphorylating its regulatory (R) domain.
- CFTR mutations cause cystic fibrosis (CF); the two common mutations, ΔF508 and G551D, reportedly also impair channel activation by PKA.
- We show here, for both mutants, that PKA-dependent activation is slowed despite intact binding affinity for the kinase, and that non-catalytic activation dominates overall channel activity and might be further enhanced by stabilization of the NBD1–NBD2–TMD interface.
- For ΔF508 CFTR, but not for G551D CFTR, a combination of clinically used potentiator drugs evokes substantial PKA-independent channel activity but suppresses non-catalytic activation by PKA.
- These findings help us to understand the activation defects caused by two common CF mutations and suggest room for further improvement of potentiator drugs currently used in CF therapy.

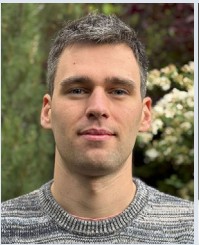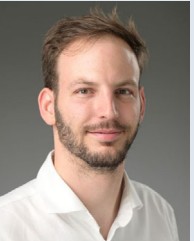

**Olivér Závoti** obtained his medical degree in 2023 and is pursuing a PhD in cellular and molecular physiology at Semmelweis University, Budapest. His research focuses on the CFTR anion channel, mutations of which cause cystic fibrosis. He has been investigating the functional consequences of multiple pathogenic CFTR mutations and the pharmacological modulation of these variants. In the course of his work, he has gained expertise in patch-clamp electrophysiology, molecular biology, cell culture, and protein biochemistry. **Marton A. Simon**, a chemist by training, earned his Ph.D. in molecular medicine at Semmelweis University, where he conducted structure-function studies of the CFTR ion channel. He then completed his postdoctoral training at The Rockefeller University, investigating the mechanism of HDL formation. Building on that experience, he now serves as a senior scientist and project manager at VRG Therapeutics, where he is involved in the development of gene therapy for focal epilepsy.

## Introduction

The cystic fibrosis transmembrane conductance regulator (CFTR) anion channel is expressed in epithelial cells of the lung, liver, pancreas, digestive tract, reproductive tract, sweat duct and kidney and localizes to the apical surfaces of these epithelia (Riordan et al., 1989). CFTR-mediated chloride and bicarbonate fluxes are crucial for normal salt–water homeostasis, and altered CFTR function leads to disease (Saint-Criq & Grey, 2017). Diminished CFTR activity, caused by pathological CFTR mutations, causes cystic fibrosis (CF) (O'Sullivan & Freedman, 2009). Excessive CFTR activity underlies secretory diarrhoea caused by bacterial toxins (Thiagarajah et al., 2015) and also cyst growth in autosomal dominant polycystic kidney disease (Jouret & Devuyst, 2020). In addition, non-CF bronchiectasis, chronic rhinosinusitis and a range of other CF-related conditions are more prevalent in CF carriers (Miller et al., 2020; Pignatti et al., 1995; Wang, Moylan et al., 2000) and in people with acquired CFTR dysfunction (Alexander et al., 2012; Clunes et al., 2012; Pallagi et al., 2024).

CFTR belongs to the family of ATP-binding cassette (ABC) proteins and comprises two transmembrane domains (TMD1 and TMD2) that form the anion pore, two cytosolic nucleotide binding domains (NBD1 and NBD2) and a unique unstructured regulatory (R) domain (Riordan et al., 1989). Upon ATP binding, the two NBDs associate into a tight head-to-tail dimer, which occludes two ATP molecules at the interface, both sandwiched between the highly conserved Walker A and B motifs of one NBD and the ABC-specific signature sequence of the other (Zhang et al., 2018). The binding site flanked by the Walker motifs of NBD2 (site 2) is catalytically active (Li et al., 1996; Ramjeesingh et al., 1999). Hydrolysis of ATP here causes loosening of the dimer interface around site 2, allowing nucleotide exchange and initiation of a new cycle. In contrast, the composite site flanked by the Walker motifs of NBD1 (site 1) is catalytically inactive (Aleksandrov et al., 2002; Basso et al., 2003) and remains ATP bound and dimerized throughout several catalytic cycles at site 2 (Levring et al., 2023; Tsai et al., 2010). Opening and closing (gating) of the anion pore follows a bursting pattern and is linked to the above NBD catalytic cycle. Initiation of a burst of openings is linked to tightening of the site 2 interface after ATP binding (Vergani et al., 2005), whereas termination of a burst is linked to loosening of the site 2 interface after ATP hydrolysis (Csanády et al., 2010; Gunderson & Kopito, 1995; Simon et al., 2023).

In live cells, CFTR activity is regulated by the catalytic subunit of cAMP-dependent protein kinase (PKA), which phosphorylates multiple serines in its R domain (Anderson et al., 1991; Cheng et al., 1991). The unphosphorylated R domain is predominantly located in a conformation wedged between the two NBDs, preventing their dimerization (Fig. 1A, state 1), and only rarely transitions into a 'released' conformation permissive to channel gating (Fig. 1A, state 2) (Liu et al., 2017). PKA activates CFTR channels by promoting R-domain release through two functionally additive mechanisms. Phosphorylation of the R domain shifts its conformational equilibrium towards released (state 5 vs. state 6; Zhang et al., 2018), causing catalytic channel activation. In addition, regardless of whether the R domain is phosphorylated or not, simple binding of PKA (Fig. 1A, orange) to CFTR further stabilizes the R domain in its released conformation (Fig. 1, states 3 and 4), causing additional, non-catalytic channel stimulation (Mihályi et al., 2020). Non-catalytic CFTR stimulation requires membrane anchoring of PKA through its N-terminal myristoyl group (Fig. 1A, purple zigzag) and is prevented by the N-terminal helix of the inhibitory peptide PKI(6–22) (Mihályi et al., 2024). In cryo-electron microscopy (cryo-EM) structures of the open CFTR channel in complex with PKA, the kinase (Fig. 1B, orange surface) binds between NBD1 and the 'lasso motif' in TMD1 of CFTR, and is approached by a C-terminal ∼30-residue segment of the R domain (residues 806–833; Fig. 1B, red surface), which is docked to the outer surface of NBD1 and the NBD1–TMD interface. When the structure of PKI(6–22) is mapped onto the complex (Fig. 1B, mesh), its N-terminal helix (Fig. 1B, dark purple mesh) clashes with the docked R-domain loop, suggesting that the latter is an essential component of the non-catalytically activated state (Fiedorczuk et al., 2024).

The pathological consequences of the >1000 CFTR mutations identified in CF patients are diverse, including defects in protein synthesis, maturation and trafficking, protein stability, channel gating or anion permeation through the open pore (De Boeck & Amaral, 2016). Thanks to recent breakthroughs in the development of potentiator compounds that enhance gating and of corrector compounds that boost surface expression of mutant CFTR channels, the quality of life of people with CF has improved greatly. In particular, the elexacaftor–tezacaftor–ivacaftor (ETI) drug combination has been approved for mutations that affect ∼90% of CF patients (Barry et al., 2021). Of its three components, all of which bind to the TMDs of CFTR (Fiedorczuk & Chen, 2022), ivacaftor (Fig. 1B, blue spheres) and elexacaftor (Fig. 1B, yellow spheres) act as potentiators (Laselva et al., 2021; Shaughnessy et al., 2021; Van Goor et al., 2009; Veit et al., 2021). Because ETI treatment only partly restores CFTR activity in treated patients, and some patients are intolerant to the drugs, the development of alternative small-molecule CFTR activators continues to be pursued actively (Mall et al., 2024).

The most common CF-causing mutation, deletion of phenylalanine 508 (ΔF508), diminishes channel surface

expression (Cheng et al., 1990) and profoundly impairs channel gating (Miki et al., 2010). From a structural point of view, in wild-type (WT) CFTR, residue F508 (Fig. 1*B*, purple spheres) contributes to the surface of NBD1 that interacts with the TMDs (Zhang & Chen, 2016), and its deletion destabilizes that interface (Fiedorczuk & Chen, 2022). From a functional point of view, openings of ΔF508 CFTR channels remain coupled to tight NBD dimerization and closures to ATP hydrolysis, because mutations that disrupt ATP hydrolysis at site 2 prolong its bursts by ~100-fold (Jih et al., 2011; Kopeikin et al., 2014). The major gating defect of the mutant is a greatly slowed opening rate (Miki et al., 2010), attributable to destabilization of the transition state for opening, a transient structure characterized by strain at the NBD–TMD interface (Sorum et al., 2015).

The relatively common CF-associated mutation G551D does not impair surface expression (Cheng et al., 1990) but severely disrupts channel gating (Bompadre et al., 2007; Li et al., 1996). Residue G551 (Fig. 1*B*, purple spheres) is located in the signature sequence of NBD1, and the negative charge of the non-native aspartate side chain in the G551D mutant repels the $\gamma$-phosphate of ATP bound in site 2. Thus, for G551D CFTR channels, ATP binding at site 1 stabilizes the NBD dimer interface, whereas binding at site 2 destabilizes it (Lin et al., 2014). Whether the pore openings of this mutant are coupled to NBD dimerization at site 2 remains unclear. A complete absence of site 2 dimerization has been suggested based on the lack of gating stimulation by ATP (Lin et al., 2014). On the contrary, the small but measurable ATPase activity of G551D CFTR (Li et al., 1996) argues that the tight dimer might form occasionally.

Most functional studies on the ΔF508 and G551D mutants have focused on understanding the impacts of these mutations on the ATP-dependent gating cycle. However, impaired PKA-dependent activation has also been reported for both ΔF508 (Drumm et al., 1991; Wang, Zeltwanger, et al., 2000) and G551D (Cui et al., 2019; Wang et al., 2020; Wilkinson et al., 1996) CFTR. The ongoing development of phosphodiesterase inhibitors that augment PKA activity by elevating cellular cAMP levels, for use as monotherapy or in combination with available CFTR modulators (de Poel et al., 2023; Della Sala et al., 2024; Liu et al., 2005; Rab et al., 2025; Turner et al., 2020), underlines the importance of better under-

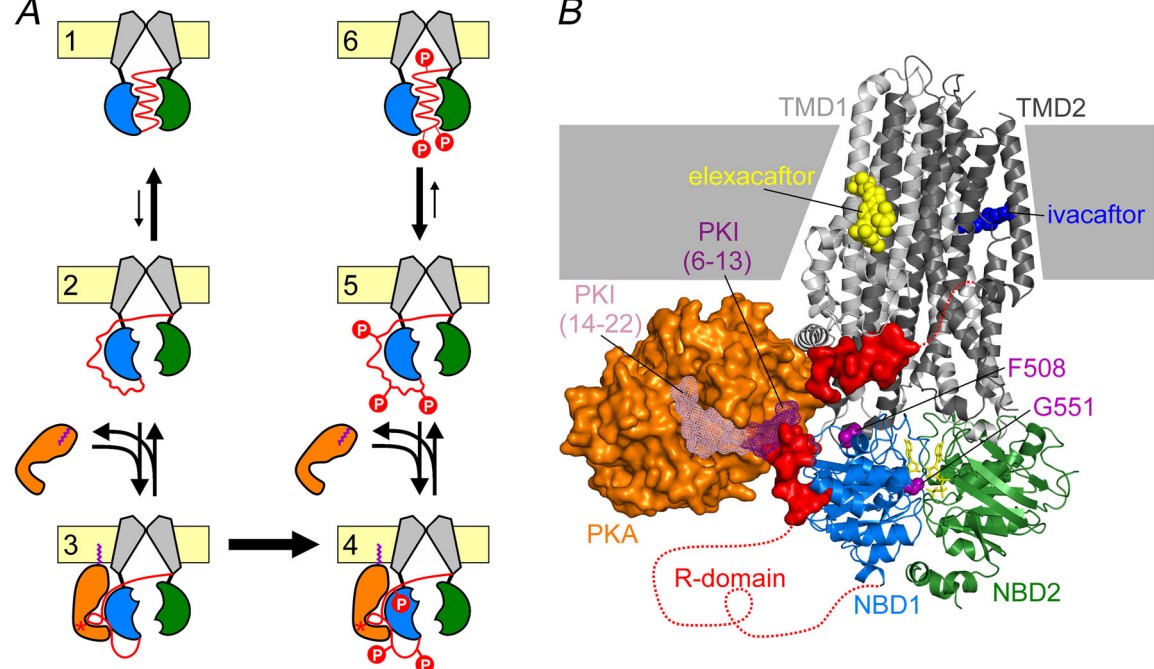

**Figure 1.  Interaction of PKA with CFTR**
*A*, simplified kinetic model of CFTR channel regulation by its R domain and PKA. Colour code: CFTR TMDs, grey; NBD1, blue; NBD2, green; R domain, red ribbon; membrane, yellow; PKA protein, orange; catalytic site, red asterisk; myristoyl group, purple zigzag. Compound states 2–5 are gating competent, that is, they allow channel gating in the presence of ATP (or P-ATP). *B*, cryo-EM structure of the complex of PKA with phosphorylated open (E1371Q) CFTR in the presence of ATP (PDBID: 9dw9). Colour coding is as in *A*. The structure of PKI(6–22) (pink/dark purple mesh) and the structures of the drugs elexacaftor (yellow spheres) and ivacaftor (blue spheres), were mapped onto the complex based on pdb entries 2gfc and 8eiq, respectively. ATP bound to NBDs of CFTR, yellow sticks. The side-chain for F508 (purple) and the backbone of G551 (purple) are shown as spheres. Red dotted lines represent unresolved R-domain segments 636–805 and 834–845 and are not drawn to scale.

standing CFTR–PKA interactions. Building on recent insights into the mechanisms of PKA-dependent CFTR channel regulation, the aim of the present study was to investigate the activation defects of ΔF508 and G551D CFTR. Specifically, we aimed to dissect mutational effects on non-catalytic *vs.* catalytic activation by PKA and to analyse how these processes are affected by clinically used CFTR modulator drugs.

## Methods

### Ethical approval

Animal experiments were approved by the Semmelweis University Animal Welfare Body (approval number: SEMMAWB/2023-001). Semmelweis University operates following the guidelines of the Hungarian Medical Research Council (Egészségügyi Tudományos Tanács). All animal research within this study adheres to the policies of *The Journal of Physiology* regarding animal experiments.

### Molecular biology

The ΔF508 CFTR/pGEMHE and G551D CFTR/pGEMHE plasmids (Csanády & Töröcsik, 2019) were linearized with Nhe I (New England Biolabs) and transcribed *in vitro* (mMessage-mMachine T7 Ultra Kit, Agilent Technologies). Purified cRNAs were stored at −80°C.

### Expression of mutant CFTR channels in *Xenopus laevis* oocytes

Mutant CFTR channels were expressed as described by Závoti and Csanády (2025). Adult female *Xenopus laevis* (RRID: NXR_0.0080) were kept at ∼18°C, under a 12 h–12 h light–dark cycle, and had free access to food. Animals were anaesthetized by submersion into a tricaine (1%) solution, and oocytes were extracted by partial ovariectomy. Animals were euthanized by freezing (−20°C) before the return of consciousness. Oocytes were digested with collagenase type II, stored at 18°C in a modified frog Ringer solution, and injected with 10–30 ng of cRNA. Recordings were made 2–4 days after injection.

### Inside-out patch-clamp recordings

Electrophysiological recordings were performed as described by Závoti and Csanády (2025). The patch pipette solution contained (mM): 138 *N*-methyl-D-glucamine (NMDG), 2 MgCl$_2$ and 5 Hepes (pH adjusted to 7.4 with HCl). The bath solution contained (mM): 138 NMDG, 2 MgCl$_2$, 5 Hepes and 0.5 EGTA (pH adjusted to 7.1 with HCl). The continuously flowing bath solution could be exchanged

with a time constant of <100 ms using electronic valves (ALA-VM8, Ala Scientific Instruments). Recordings were obtained at 25°C; membrane potential was −40 mV for macroscopic recordings and −80 mV for single-channel recordings. Currents were low-pass filtered at 1 kHz, amplified (Axopatch 200B, Molecular Devices), digitized at 10 kHz (Digidata 1550B, Molecular Devices), and recorded to disc (pCLAMP 11, Molecular Devices; RRID: SCR_01 1323). Stock solutions of concentrations >100× for MgATP (Sigma, aqueous, pH adjusted to 7.1 with NMDG), $N^6$-(2-phenylethyl)-ATP (P-ATP; Biolog LSI, aqueous), ivacaftor (Selleck Chemicals, DMSO-based) and elexacaftor (Selleck Chemicals, DMSO-based) were used for dilution into the bath solution. The catalytic subunit of bovine protein kinase A (PKA) was prepared from beef heart as described by Mihályi et al. (2024) and was diluted into the bath solution from 40–60 μM stock solutions. For the addition of 1.2 μM PKA (Fig. 7), the ∼150 mM potassium phosphate buffer of the PKA stock was exchanged to 10 mM potassium phosphate using a Hi-TRAP desalting column (GE Healthcare). Thus, in all experiments, the final phosphate concentration after PKA addition remained below ∼1 mM.

### Data analysis

Macroscopic raw data traces were Gaussian filtered at 50 Hz, and the baseline was subtracted. Fractional steady-state currents in various test conditions were obtained by dividing the mean current in the test condition by that in a reference condition (e.g. ATP + PKA or P-ATP + PKA) in the same patch.

For single-channel kinetic analysis (Figs 3 and 9), current traces with up to seven superimposed channel openings were Gaussian filtered at 50 Hz, idealized by half-amplitude threshold crossing, and the open probability ($P_o$) calculated from the event lists as: $P_o = (\sum_k t_k l_k)/(NT)$, where $t_k$ and $l_k$ denote the duration and current level, respectively, of the $k$th event, $N$ is the number of channels in the patch, and $T$ is the total recording time. The value of $N$ was estimated as the maximum number of simultaneously open channels in the patch in the presence of ATP + PKA + drugs; because the true number of channels might be underestimated, this approach yields an upper estimate of $P_o$. In the absence of drugs (Fig. 3), $P_o$ is very small, hence $N$ is heavily under-estimated, and in such conditions only relative $P_o$ values are given, normalized to that in the presence of ATP + PKA in the same patch (Fig. 3B–D, left). Mean burst ($T_b$) and mean interburst ($T_{ib}$) durations were calculated as described, by simultaneous maximum likelihood fitting of the dwell time distributions at all conductance levels, with correction for an imposed fixed dead time of 6 ms (Csanády, 2000). Given that the estimated value of $T_{ib}$ depends heavily on the correct estimation of $N$, only

relative $T_{ib}$ values are given, normalized to that in the presence of ATP + PKA in the same patch (Fig. 3B–D, right).

### Statistics

All bar graphs represent the mean ± SD, with the numbers of independent experiments (individual patches) provided in the figure legends. In addition, individual data points are also depicted. Statistical significances were evaluated using Student's *t* test, with a threshold set to $P < 0.05$. Exact *P* values are provided in the figure legends.

## Results

### High-affinity ATP analogue enhances non-catalytic stimulation by PKA for both ΔF508 and G551D CFTR

The ATP analogue $N^6$-(2-phenylethyl)-ATP (P-ATP) binds with high affinity to the NBDs of CFTR and efficiently drives CFTR channel gating (Csanády et al., 2013; Zhou et al., 2005), but it does not bind to PKA and cannot be used for phosphotransfer (Mihályi et al., 2020; Schauble et al., 2007). Non-catalytic CFTR activation by PKA binding can therefore be estimated conveniently by applying the kinase to channels that are gating in P-ATP (Mihályi et al., 2020). To gauge the extent of non-catalytic activation by PKA of non-phosphorylated and phosphorylated ΔF508 and G551D CFTR, inside-out patches excised from *X. laevis* oocytes expressing human CFTR channels with the above mutations were exposed repeatedly for ~1 min intervals to 300 nM PKA, initially in the presence of P-ATP (10 μM), then in the presence of ATP (2 mM), and finally in the presence of P-ATP again (Fig. 2A and B). The bracketing PKA exposures in P-ATP selectively evoke non-catalytic stimulation of unphosphorylated and phosphorylated channels, respectively, by PKA. In contrast, the second exposure, in ATP, causes both catalytic stimulation (i.e. phosphorylation) and non-catalytic stimulation; the latter component is revealed by the sudden partial current decline immediately after PKA removal (Fig. 2A and B, centre; Fig. 2C and D, second blue double-headed arrow) (Mihályi et al., 2020). Of note, in our inside-out patches, non-catalytic activation is rapidly reversible upon PKA washout, whereas catalytic activation is irreversible owing to the absence of cytosolic phosphatases, which dephosphorylate CFTR in intact cells (Mihályi et al., 2020).

For both mutants, non-catalytic activation was clearly observable both before and after phosphorylation. However, the amplitude of non-catalytic current activation of phosphorylated channels by PKA was markedly enhanced in the presence of P-ATP compared with that in ATP (Fig. 2C and D, compare second and third blue double-headed arrow): by >2-fold for ΔF508

and by ~10-fold for G551D CFTR. Moreover, for ΔF508 CFTR, non-catalytic stimulation was ~2-fold larger even for non-phosphorylated channels in P-ATP than for phosphorylated channels in ATP (Fig. 2C, compare first and second blue double-headed arrow). These findings are in contrast to WT CFTR, for which the amplitude of the non-catalytic current activation is little affected by the choice of nucleotide (ATP or P-ATP) used to drive channel gating (Fig. 2E, compare blue double-headed arrows; data replotted from Mihályi et al., 2020).

### Non-catalytic activation by PKA modulates single-channel gating parameters of ΔF508 and G551D CFTR in a qualitatively similar manner to those of WT CFTR

To verify whether non-catalytic stimulation of ΔF508 and G551D CFTR channels by PKA obeys a mechanism similar to that described for WT CFTR, we used patches with small numbers of channels, in which individual gating transitions could be resolved clearly, to estimate single-channel gating parameters for both mutants in the various experimental conditions shown in Fig. 2 (Fig. 3A and C). Given that steady-state kinetic analysis requires longer segments of record, each experimental segment was prolonged to ~2 min. This type of analysis is challenging experimentally, owing to the large dynamic range of $P_o$ values that the channels sample during the protocol. In patches in which unitary transitions of phosphorylated channels remained resolvable, we typically did not observe any openings for unphosphorylated channels gating in ATP or P-ATP alone. Thus, for those conditions, kinetic analysis was not performed. Furthermore, over the prolonged time course of these experiments, ΔF508 channels already started to show signs of rundown. Despite these limitations, the obtained estimates of normalized $P_o$ values (Fig. 3B and D, left) are roughly consistent with the patterns of fractional currents observed in macroscopic patches (Fig. 2C and D). As described earlier for WT CFTR (Csanády et al., 2010), for both mutants the loss of non-catalytic stimulation upon PKA removal was associated with a large decrease in mean burst duration ($T_b$; Fig. 3B and D, centre, compare light grey with preceding red bar) and a more modest increase in mean interburst duration ($T_{ib}$; Fig. 3B and D, right, compare light grey with preceding red bar), although quantification of $T_{ib}$ was subject to technical limitations owing to the unknown true number of channels in the patch (see Methods section). These findings suggest that PKA reversibly stimulates both mutants through a mechanism similar to that described for WT CFTR. In addition, replacing ATP with P-ATP strongly prolonged mean burst durations of phosphorylated G551D CFTR channels not only in the

absence (cf. Bompadre et al., 2008), but also in the presence of PKA (Fig. 3D, centre, compare red bars). For phosphorylated ΔF508 CFTR channels, such an effect was not observed, possibly owing to progressive, time-dependent rundown (Fig. 3B, centre, compare red bars).

### ΔF508 and G551D CFTR currents evoked by PKA develop more slowly than WT and are dominated by non-catalytic activation

For both ΔF508 (Wang, Zeltwanger et al., 2000) and G551D (Cui et al., 2019) CFTR, current activation rate

in the presence of ATP + PKA was reportedly slowed, suggesting that the 1 min exposure to ATP + PKA (see Fig. 2A and B, centre) might be insufficient to reach full phosphorylation. We therefore studied, for both mutants, current activation over a prolonged, 4 min PKA exposure time (Fig. 4A and D). Consistent with the earlier report (Wang, Zeltwanger et al., 2000), current development was slowed for ΔF508 CFTR; after 1 min in ATP + PKA, the current reached only ~50% of its maximal value (Fig. 4A and G, red bars), in contrast to WT CFTR for which ~90% of the current developed within the first minute (Fig. 4I, red bars; data replotted from Mihályi et al., 2024). For G551D CFTR, a similar analysis (Fig. 4H) did not reveal

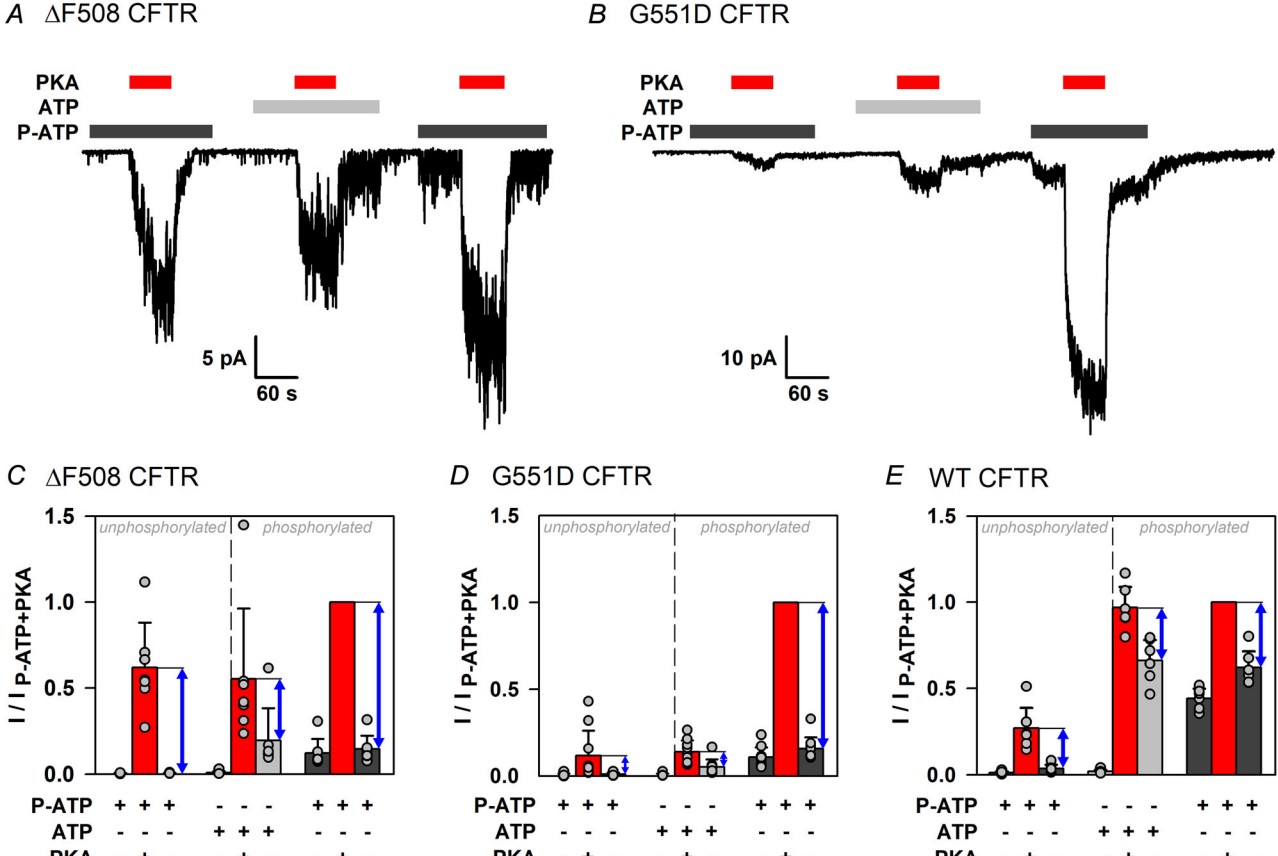

**Figure 2. For both ΔF508 and G551D CFTR, non-catalytic activation by PKA is enhanced in the presence of P-ATP**

*A and B, current recordings from inside-out patches excised from Xenopus laevis oocytes expressing ΔF508 (A) or G551D (B) CFTR. Channel activity was evoked by cytosolic exposures to 10 μM P-ATP (dark grey bars) or 2 mM ATP (light grey bars), with or without 300 nM PKA (red bars). Membrane potential was −40 mV, temperature was 25°C. C–E, mean steady-state currents in the nine consecutive segments of the experimental protocol shown in A and B, for ΔF508 (C), G551D (D) and WT (E) CFTR, normalized to the mean current observed during the third PKA exposure in the same patch. Data in E are replotted from fig. 6B of Mihályi et al. (2020) but have been renormalized. Bars show the mean ± SD, n = 7 (C), n = 10 (D) and n = 6 (E). Blue double-headed arrows represent the magnitude of non-catalytic CFTR stimulation by PKA for unphosphorylated channels in P-ATP and for (partly) phosphorylated channels in ATP or P-ATP, respectively. The ratio of the amplitudes of non-catalytic stimulation of phosphorylated channels in P-ATP vs. in ATP (cf. C–E, ratio of third to second blue arrow), calculated individually in each patch, was 3.39 ± 2.01 (n = 7) for ΔF508, 12.5 ± 5.0 (n = 10) for G551D and 1.27 ± 0.42 (n = 6) for WT CFTR [mean ± SD (n)]. The ratio was significantly larger for both mutants compared with WT (P = 0.0285 and P = 9.06 × 10^{-5}, respectively, Student's paired t test).*

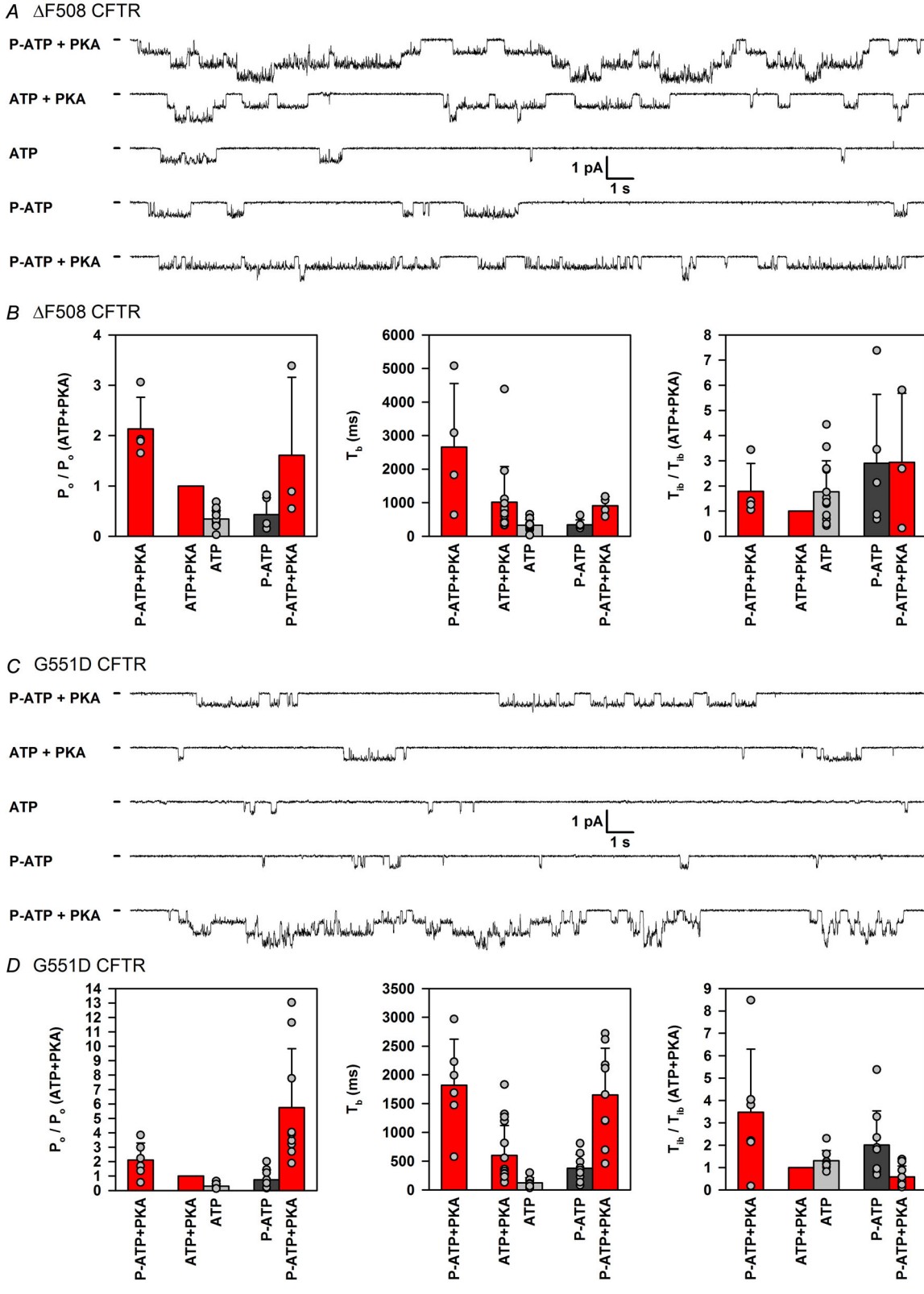

**Figure 3. Non-catalytic activation by PKA modulates single-channel gating parameters of ΔF508 and G551D CFTR a qualitatively similar manner to those of WT CFTR**

*A* and *C*, segments of current recording from inside-out patches containing small numbers of ΔF508 (*A*) or G551D (*C*) CFTR channels, following the experimental protocol in Fig. 2*A* and *B*, but allowing for longer (∼2 min) observation times in each experimental segment. Traces shown represent final stretches of consecutive segments

of the same patches. Top two traces illustrate gating in P-ATP (10 µM) + PKA (300 nM) prior to phosphorylation and in ATP (2 mM) + PKA. Bottom three traces illustrate gating following phosphorylation, in ATP, in P-ATP and in P-ATP + PKA, respectively. Dashes before current traces mark zero-current levels. Traces in ATP or P-ATP prior to phosphorylation are not shown, because in patches with such low channel numbers, typically no channel openings were observed. *B* and *D*, normalized open probability ($P_o$; left panels), mean burst duration ($T_b$; centre panels) and normalized mean interburst duration ($T_{ib}$; right panels) for ΔF508 (*B*) or G551D (*D*) CFTR channels in the five experimental conditions illustrated by the current traces. Given that the number of active channels could not be determined, $P_o$ and $T_{ib}$ are shown normalized to the respective parameter in the presence of ATP + PKA in the same patch (see Methods section). Bars show the mean ± SD, $n = 4$–15 (*B*) and $n = 6$–16 (*D*).

reduced fractional activation after 1 min in PKA, but the large stochastic fluctuations of the small currents of this mutant limit the applicability of this type of analysis. To overcome the stochastic nature of individual gating events, we synthesized, for both mutants, macroscopic current activation time courses by summing the currents of eight patches synchronized to the time points of PKA addition (Fig. 4*B* and *E*). These summed macroscopic currents afforded estimation of the activation half-time ($t_{\frac{1}{2}}$), which was ~40 s for ΔF508 and ~29 s for G551D CFTR (Fig. 4*B* and *E*; L-bars and black numbers), that is, both indeed prolonged relative to that of WT CFTR [$t_{\frac{1}{2}} = 11.8 ± 1.4$ s ($n = 13$); cf. Fig. 4*B* and *E*, blue WT current trace].

Interestingly, the relative contributions of the reversible (non-catalytically activated) *vs.* irreversible (catalytically activated) component to the total current were also altered in both mutants. For phosphorylated WT CFTR in the presence of 300 nM PKA, the reversible and irreversible components contributed roughly equally to the total current (Fig. 4*I*, right; compare blue double-headed arrow with grey bar), whereas for phosphorylated ΔF508 and G551D channels the total current in the presence of ATP + PKA was dominated by the reversible component (Fig. 4*G* and *H*, right). Synthesized macroscopic currents, obtained by summing currents from multiple patches synchronized to the time points of PKA removal (Fig. 4*C* and *F*), suggest that, for both mutants, the reversible component accounts for up to ~75–80%, and the irreversible component for only ~20–25%, of the total current in the presence of ATP + 300 nM PKA. Thus, both mutations compromise catalytic activation more severely than non-catalytic activation. Of note, the summed trace for G551D (Fig. 4*F*) reproduces the reported biphasic current response to ATP removal, caused by rapid washout of inhibitory ATP (from site 2) followed by slower loss of stimulatory ATP (from site 1) (Lin et al., 2014).

Although the data in Fig. 4*B* and *G* suggest that ΔF508 channels did not become fully phosphorylated in the experimental protocol shown in Fig. 2*A*, the strong enhancement of non-catalytic stimulation in the presence of P-ATP was readily confirmed with longer PKA exposures (>2 min in P-ATP, ~4 min in ATP; Fig. 5*A*). Because, over such long experimental time courses, in some patches substantial inactivation was apparent, the currents in the three subsequent sections (P-ATP, ATP and P-ATP) of these experiments were analysed separately,

each normalized to that in the presence of PKA in the respective section. Strong enhancement of non-catalytic stimulation of phosphorylated ΔF508 channels by PKA in P-ATP *vs.* ATP (cf. Fig. 2*A* and *C*) was again revealed by larger fractional stimulation (Fig. 5*B*, right *vs.* centre; ratio of second to third bar). Domination of total ΔF508 channel activity in ATP + PKA by non-catalytic stimulation (cf. Fig. 4*A* and *G*) was also evident from the larger fractional current reduction upon PKA removal compared with WT (Fig. 5*B* vs. *C*, centre, third bar).

### Both non-catalytic and catalytic stimulation by PKA is slowed by the ΔF508 mutation

To understand which component (non-catalytic or catalytic) is responsible for the slowed current activation of ΔF508 CFTR (Fig. 4*B* and *G*), we dissected the contribution of each component to the total current at various time points of the activation process. To that end, 300 nM PKA was added initially for 1 min, then for an additional 3 min to reach full phosphorylation (Fig. 6*A*). Upon PKA removal, the surviving current reveals the size of the irreversible (catalytic) component at that point in time. This analysis (Fig. 6*B*) revealed that for ΔF508 CFTR, after 1 min in ATP + 300 nM PKA, the reversible (non-catalytic) component reaches only ~50% (Fig. 6*B*, blue double-arrows) and the irreversible component only ~45% (Fig. 6*B*, grey bars) of its maximal value. In contrast, for WT CFTR, the reversible component develops fully (Fig. 6*C*, blue double-arrows), and the irreversible component reaches ~70% of its maximal value (Fig. 6*C*, grey bars), after 1 min in ATP + 300 nM PKA (Mihályi et al., 2024). Thus, the slowed overall activation rate of the mutant, compared with WT CFTR, is caused by a slower development of both components.

### The affinity for PKA binding is intact for both ΔF508 and G551D CFTR

In an earlier report, a right shift in the dose–response curve for G551D current activation by PKA was interpreted to reflect a decreased affinity of the mutant for the kinase (Wang et al., 2020). To evaluate whether such a mechanism might explain the slowed activation time courses of the two mutants (Fig. 4*B* and *E*), we exploited reversible (non-catalytic) activation to gauge

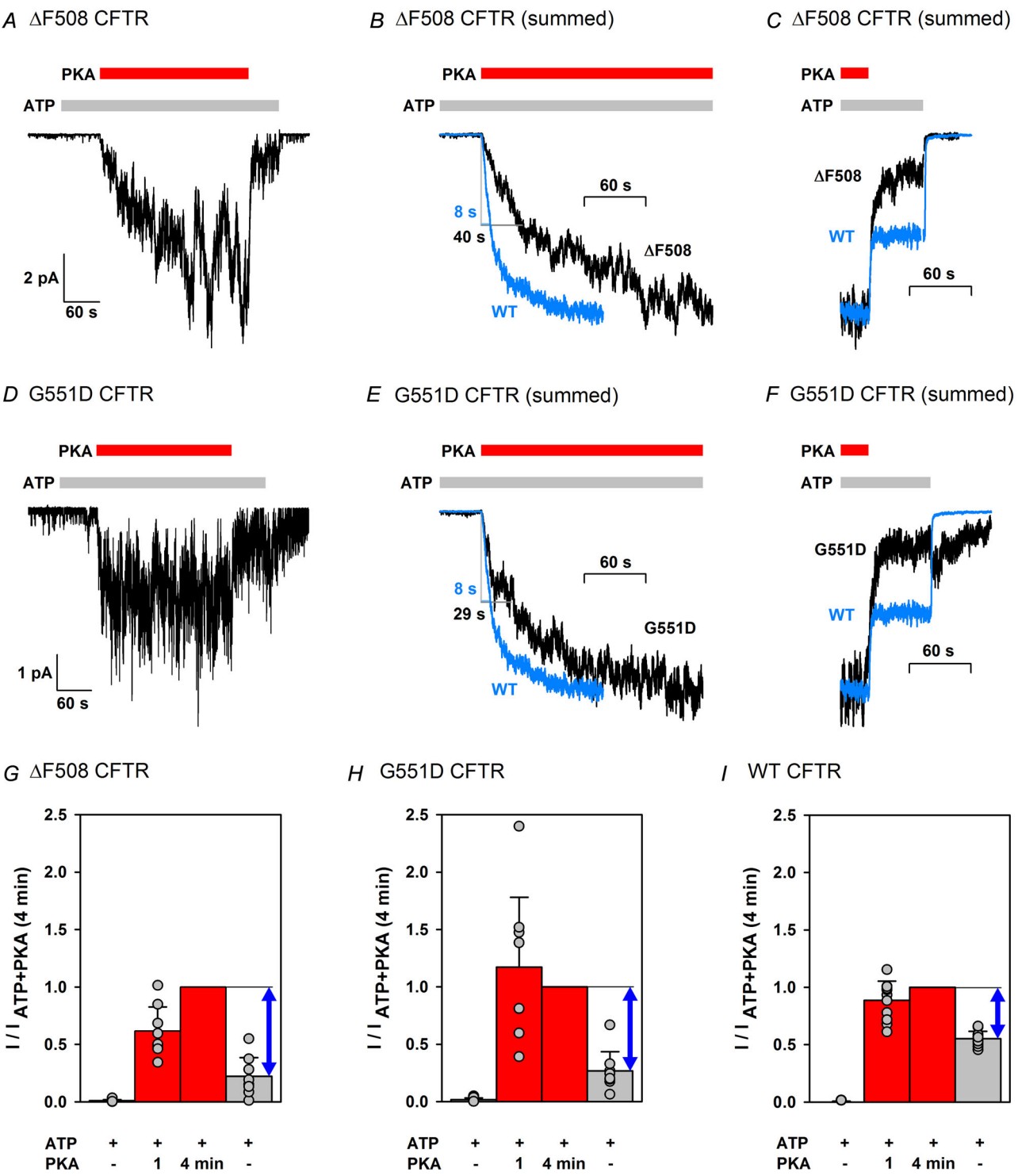

**Figure 4. ΔF508 and G551D CFTR currents evoked by PKA develop slower than WT and are dominated by non-catalytic activation**

*A* and *D*, quasi-macroscopic inside-out patch recordings of ΔF508 (*A*) and G551D (*D*) channels in response to a 4 min exposure to 300 nM PKA (red bars) in the presence of 2 mM ATP (grey bars). *B*, *C*, *E* and *F*, synthesized macroscopic currents obtained by summing quasi-macroscopic currents from 6 to 11 patches, illustrating channel activation upon addition of PKA (*B* and *E*) and (partial deactivation upon PKA removal *C* and *F*), for both ΔF508 (*B* and *C*) and G551D (*E* and *F*) CFTR. Individual current traces were synchronized to the time point of PKA addition (*B* and *E*) or PKA removal (*C* and *F*). Overlaid blue traces illustrate the respective current time courses for a single representative macroscopic WT patch. All traces were normalized to their maximal values. In *B* and *E*, L-shaped bars and numbers depict activation half-times ($t_{\frac{1}{2}}$, in seconds). *G–I*, fractional currents in the presence of 2 mM

ATP before PKA exposure (first bar), after 1 min (second bar) or 4 min (third bar) of exposure to 300 nM PKA (averaged over 20 s), and following PKA removal (fourth bar) for ΔF508 (*G*), G551D (*H*) and WT (*I*) CFTR. Currents were normalized to those after 4 min in PKA in the same patch. Bars show the mean ± SD, *n* = 8 (*G*), *n* = 8 (*H*) and *n* = 11 (*I*). Blue double-headed arrows represent the magnitude of non-catalytic CFTR stimulation by PKA after a 4 min PKA exposure. Data in *I* are replotted from Fig. 1*L* of Mihályi et al. (2024). The ratio of the current amplitude in ATP following PKA removal to that in ATP + PKA (Fig. 4*G–I*, fourth bar) was significantly smaller for both ΔF508 and G551D CFTR compared with WT (*P* = 1.08 × 10$^{-5}$ and *P* = 7.90 × 10$^{-5}$, respectively, Student's paired *t* test).

their apparent affinities for PKA binding. To that end, phosphorylated ΔF508 and G551D channels gating in 2 mM ATP were exposed sequentially to 300 nM and 1.2 μM PKA (Fig. 7*A* and *B*), and the amplitudes of reversible stimulation by the two concentrations of PKA were compared within each patch (Fig. 7*C* and *D*). For both mutants, the 4-fold higher PKA concentration increased the amplitude of the reversible current component by only ∼50% (Fig. 7*C* and *D*; compare two blue double-headed arrows), suggesting that the $K_d$ for PKA binding is <300 nM. This estimate is comparable to that obtained earlier for WT CFTR (Fiedorczuk et al., 2024), suggesting that the affinity for PKA binding is not impaired in either mutant.

### In the presence of elexacaftor + ivacaftor, non-catalytic activation by PKA is undetectable for phosphorylated ΔF508 CFTR

Most CF patients who carry ΔF508 alleles are currently treated with ETI. We tested how the combination of the two potentiator constituents (elexacaftor + ivacaftor) affects ΔF508 channel activation by PKA. Exposure of unphosphorylated ΔF508 CFTR channels to 10 μM P-ATP in the presence of elexacaftor (1 μM) + ivacaftor (10 nM) evoked macroscopic currents, suggesting substantial channel activation by the drugs even prior to channel phosphorylation (Fig. 8*A*, left; cf. Fig. 2*A*, left). A subsequent 1 min exposure to 300 nM PKA robustly stimulated these currents, reporting strong non-catalytic activation of unphosphorylated ΔF508 CFTR channels even in the presence of the drugs (Fig. 8*A* and *B*). We next tested the effects of PKA exposure in the presence of ATP + elexacaftor + ivacaftor (Fig. 8*C* and *D*). As observed in the presence of P-ATP, exposure of unphosphorylated ΔF508 CFTR channels to ATP + drugs activated sizeable macroscopic currents, consistent with substantial activation of unphosphorylated channels by the drugs. These currents were further enhanced by only ∼2- to 3-fold upon full activation by a 4 min exposure to 300 nM PKA (Fig. 8*C* and *D*, second *vs.* first bar). Surprisingly, subsequent removal of the kinase, in the

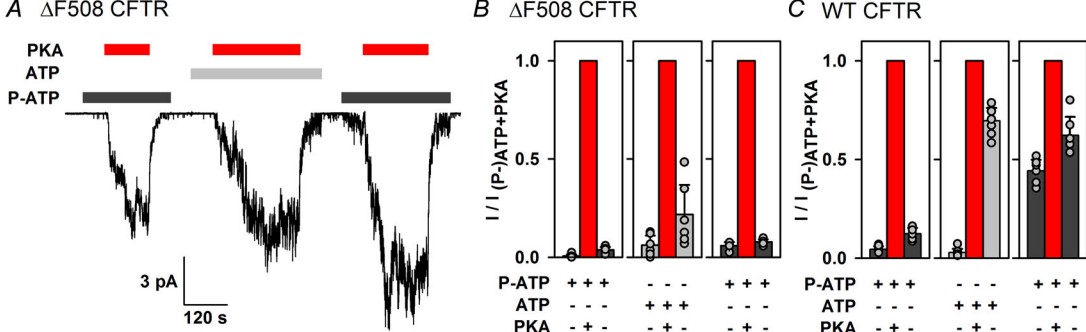

**Figure 5. Non-catalytic activation by PKA dominates, and is further enhanced in the presence of P-ATP, even for fully phosphorylated ΔF508 CFTR**
*A*, current recording from an inside-out patch expressing ΔF508 CFTR using the experimental protocol shown in Fig. 2*A*, with longer exposures to PKA. The PKA was applied for >2 min in the presence of P-ATP and for ∼4 min in the presence of ATP. *B*, fractional steady-state currents in the nine consecutive segments of the experimental protocol shown in *A*. Given that, in some patches, substantial inactivation was apparent over the time course of this long (∼18 min) experimental protocol, currents in the three subsequent sections (P-ATP, ATP and P-ATP) of the experiment are shown separately, each normalized to that in the presence of PKA in the respective section. Bars show the mean ± SD, *n* = 5–6. *C*, data for WT CFTR replotted from Fig. 2*E*, but also renormalized. For phosphorylated ΔF508 CFTR, the ratio of the current amplitudes in the presence of PKA over that following PKA removal (cf. *B*, centre and right, ratio of second to third bar), calculated individually in each patch, was 7.20 ± 4.33 (*n* = 6) in the presence of ATP, and 13.3 ± 2.5 (*n* = 5) in the presence of P-ATP [mean ± SD (*n*)]. The ratio was significantly larger in P-ATP than in ATP (*P* = 0.0223, Student's paired *t* test). The ratio of the current amplitude in ATP following PKA removal to that in ATP + PKA (*B* and *C*, centre, third bar) was significantly smaller for ΔF508 compared with WT (*P* = 1.02 × 10$^{-5}$, Student's paired *t* test).

maintained presence of the drugs, was not accompanied by a significant current decline, reporting near-complete absence of non-catalytic stimulation for phosphorylated ΔF508 CFTR channels in the presence of the drugs (Fig. 8*C*, right; Fig. 8*D*, third *vs.* second bar).

One possibility for the lack of reversible stimulation might be if binding of either ivacaftor or elexacaftor prevents PKA binding to the docking site on CFTR observed in structures (Fig. 1*B*). Such an effect cannot be attributed to ivacaftor, because non-catalytic stimulation of phosphorylated ΔF508 channels in the presence of that drug was demonstrated in an earlier study (Fig. 8*E*; replotted from Mihályi et al., 2024). We therefore evaluated channel stimulation by PKA in the presence of elexacaftor alone (Fig. 8*F* and *G*). Interestingly, in the presence of elexacaftor alone, the PKA dependence of ΔF508 channel activity was similar to that reported earlier for ivacaftor alone. No substantial currents were evoked by exposure of unphosphorylated ΔF508 CFTR channels to elexacaftor + ATP (Fig. 8*F*, left; Fig. 8*G*, first bar). Furthermore, following full channel activation by a 3 min exposure to PKA, removal of the kinase was accompanied by a rapid robust current decline (Fig. 8*F*, right; Fig. 8*G*, third *vs.* second bar). Thus, non-catalytic stimulation of phosphorylated ΔF508 channels is preserved in the presence of either ivacaftor or elexacaftor, but not when both drugs are applied simultaneously.

Lack of non-catalytic stimulation of ΔF508 CFTR by PKA in the presence of ATP + ivacaftor + elexacaftor was also confirmed in recordings with small numbers of active channels suitable for dwell-time analysis (Fig. 9*A*). The estimated open probability of phosphorylated channels in ATP + drugs remained ∼0.5, regardless of the absence or presence of PKA (Fig. 9*A* and *B*),

## Elexacaftor + ivacaftor does not compromise non-catalytic activation by PKA of phosphorylated G551D CFTR

Although most patients who carry a G551D allele receive treatment with ivacaftor alone, some patients (especially those who carry a ΔF508 allele in trans) have transitioned to ETI treatment. In intact cells, elexacaftor + ivacaftor has been reported to enhance forskolin-stimulated activity of G551D CFTR channels by two orders of magnitude (Veit et al., 2021). To obtain mechanistic insight into this process, we assessed how the presence of elexacaftor + ivacaftor affects non-catalytic *vs.* catalytic stimulation of G551D CFTR by PKA. When applied to unphosphorylated channels gating in P-ATP in the presence of elexacaftor + ivacaftor, PKA induced strong reversible stimulation (Fig. 10*A* and *B*). Furthermore, currents of unphosphorylated channels gating in ATP in the presence of elexacaftor + ivacaftor were stimulated robustly by PKA, suggesting that the $P_o$ of unphosphorylated G551D CFTR remains small even in the presence of both drugs (Fig. 10*C*, left; Fig. 10*D*, first *vs.* second bar), unlike our observations on ΔF508 CFTR (Fig. 8*C*, left; Fig. 8*D*, first *vs.* second bar). Finally, PKA removal revealed that ∼50% of the total activity in

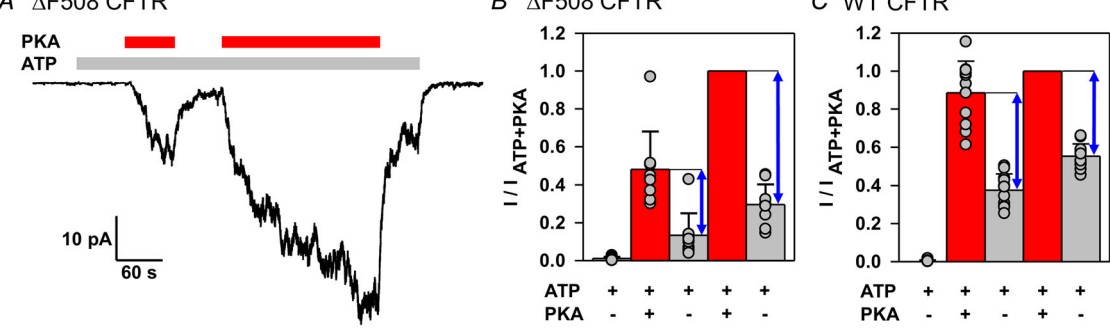

**Figure 6. Both non-catalytic and catalytic activation by PKA develop more slowly for ΔF508 CFTR**
*A*, inside-out macropatch recording of ΔF508 CFTR current evoked by consecutive 1 and 3 min exposures to 300 nM PKA (red bars), in the presence of 2 mM ATP (light grey bar). *B* and *C*, mean currents calculated over the final 20 s of each of the five consecutive segments of the experimental protocol shown in *A*, for ΔF508 (B) and WT (C) CFTR, normalized to that of the second PKA exposure in the same patch. Data in *C* are replotted from Fig. 1*L* of Mihályi et al. (2024). Bars show the mean ± SD, n = 7 (*B*) and n = 11 (*C*). Blue double-headed arrows represent the magnitude of non-catalytic CFTR stimulation by PKA after a 1 min PKA exposure, and following full phosphorylation, respectively. The ratio of the amplitudes of non-catalytic stimulation after 1 *vs.* 4 min in PKA + ATP (cf. *B* and *C*, ratio of first to second blue arrow), calculated individually in each patch, was 0.51 ± 0.21 (n = 8) for ΔF508 and 1.16 ± 0.26 (n = 11) for WT CFTR [mean ± SD (n)]. The ratio was significantly smaller for ΔF508 compared with WT (P = 2.09 × 10⁻⁵, Student's paired *t* test). The ratio of the amplitudes of catalytic stimulation after 1 *vs.* 4 min in PKA+ATP (cf. *B* and *C*, ratio of second to third grey bar), calculated individually in each patch, was 0.44 ± 0.25 (n = 8) for ΔF508 and 0.68 ± 0.13 (n = 11) for WT CFTR [mean ± SD (n)]. The ratio was significantly smaller for ΔF508 compared with WT (P = 0.0128, Student's paired *t* test).

the presence of ATP + PKA + drugs was attributable to reversible, non-catalytic stimulation (Fig. 10*C*, right; Fig. 10*D*, second *vs.* third bar). Thus, the elexacaftor + ivacaftor combination preserves and enhances both non-catalytic and catalytic activation of G551D CFTR channels by PKA.

## Discussion

For both $\Delta$F508 and G551D CFTR, the $P_o$ of phosphorylated channels in the presence of ATP + PKA is orders of magnitude lower than for WT (Bompadre et al., 2007; Jih et al., 2011; Miki et al., 2010). This necessarily implies that both the non-catalytic and the catalytic components of PKA-induced channel activation, each of which accounts for ~50% of the total activity of WT CFTR (Mihályi et al., 2024), must be compromised in the mutants. Here, we investigated how these two components of PKA-induced channel activity are affected

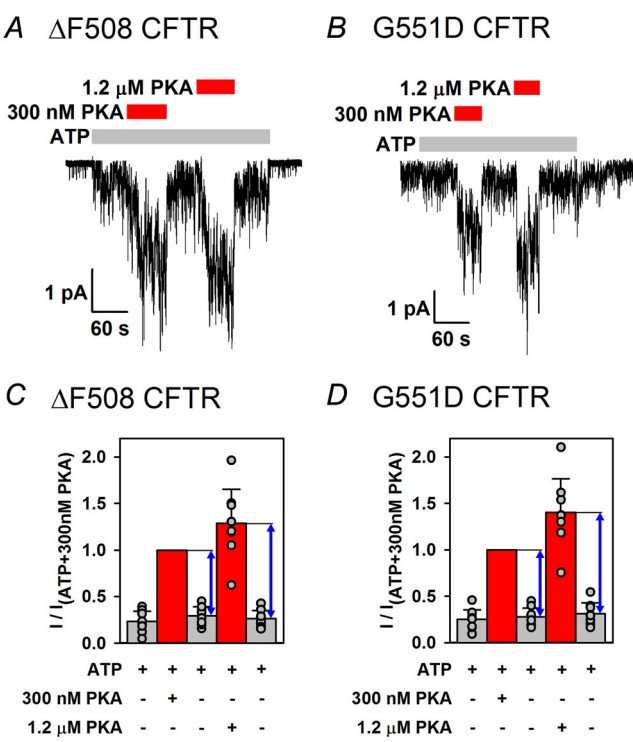

**Figure 7. Affinity for PKA binding is intact for both $\Delta$F508 and G551D CFTR**
*A* and *B*, quasi-macroscopic currents of phosphorylated $\Delta$F508 (*A*) and G551D (*B*) CFTR channels in 2 mM ATP (grey bars), and non-catalytic stimulation by 300 nM or 1.2 $\mu$M PKA (red bars). *C* and *D*, mean steady-state currents in the five consecutive segments of the experimental protocol shown in *A* and *B*, for $\Delta$F508 (*C*) and G551D (*D*) CFTR, normalized to that in 300 nM PKA in the same patch. Bars show the mean ± SD, *n* = 8 (*C*) and *n* = 8 (*D*). Blue double-headed arrows represent the magnitude of non-catalytic CFTR stimulation by 300 nM or 1.2 $\mu$M PKA.

by the two mutations. We found that both components are detectable for both mutants (Fig. 2) and are likely to obey mechanisms similar to those in WT CFTR channels (Fig. 3). However, for both mutants, overall channel activity is dominated by the non-catalytic component, which accounts for ~75% of the total current observed in the presence of ATP + PKA (Fig. 4*C* and *F–H*). Thus, both mutations impair catalytic activation more profoundly than non-catalytic activation; that is, in terms of the simplified kinetic scheme (Fig. 1*A*), the $P_o$ for compound states 2 and 5 decreases disproportionately more than that of compound states 3 and 4.

In addition, the current activation time course upon exposure to PKA + ATP was modestly slowed for G551D (Fig. 4*E*) and substantially slowed for $\Delta$F508 (Fig. 4*B*) CFTR channels, consistent with earlier reports (Cui et al., 2019; Wang, Zeltwanger et al., 2000). That slowed activation reflected (at least for $\Delta$F508 CFTR) slower development of both components of channel activity (Fig. 6). In terms of the kinetic scheme (Fig. 1*A*), that phenomenon might be explained by assuming a reduced equilibrium constant for PKA binding (steps 2→3 and 5→4), as has been suggested for the G551D mutant in an earlier study (Wang et al., 2020). However, the apparent affinities for PKA binding, as reported by the amplitude of non-catalytic current activation, were not reduced for either mutant compared with that of the WT channel (Fig. 7; cf. Fiedorczuk et al., 2024). Steps 2→3 and 5→4 in the simplified scheme (Fig. 1*A*) are likely to be compound steps that involve binding of the kinase at the NBD1–TMD interface, followed by the conformational change that leads to channel stimulation. From a kinetic point of view, our data suggest that it is the latter step, not kinase binding *per se*, that is slowed by both mutations.

Based on the scheme (Fig. 1*A*), in which catalytic activation (step 3→4) reflects R-domain phosphorylation, the slowed development of the irreversible current component for $\Delta$F508 CFTR would suggest that the mutant is phosphorylated more slowly (cf. Pasyk et al., 2015). It is formally possible that catalytic activation is not rate limited by phosphorylation *per se*, but by the conformational changes induced by phosphorylation, and that the mutation slows those latter steps. However, if so, then following a short (1 min) PKA application, which activates the irreversible current component only submaximally, that component (which survives sudden PKA removal) should continue to rise, revealing slow completion of those conformational steps. Such behaviour was not observed for either WT (Mihályi et al., 2024) or $\Delta$F508 CFTR (Fig. 5*A*), suggesting that activation of the catalytic current component is indeed rate limited by phosphorylation for both WT and the mutant.

For both mutants, earlier studies had reported stimulation of the irreversible (catalytic) component by P-ATP; that is, an increased $P_o$ of phosphorylated

channels gating in the absence of PKA (Bompadre et al., 2008; Miki et al., 2010). That effect was also observed here for G551D (Fig. 2*B* and *D*, right, light *vs.* dark grey bars), but not for ΔF508 CFTR (Fig. 2*A* and *C*, right, light *vs.* dark grey bars). The reason for the latter discrepancy is unclear but might reflect differences in experimental protocols: the 1 min ATP removal here might have caused substantial inactivation (Yeh et al., 2021). Alternatively, in our experimental conditions, the $K_{\frac{1}{2}}$ for gating stimulation of the mutants might be slightly higher than reported in the earlier studies (∼6 µM in the study by Bompadre et al., 2008), hence 10 µM P-ATP might have remained subsaturating. In contrast, we found here that, for both mutants, non-catalytic stimulation by PKA is greatly enhanced when channel gating is driven by P-ATP instead of ATP (Fig. 2): that enhancement is ∼2-fold for ΔF508 CFTR (Fig. 2*A* and *C*, blue double-headed arrows) and ∼10-fold for G551D

CFTR (Fig. 2*B* and *D*, blue double-headed arrows). Thus, P-ATP alleviates not only the gating defects but also the activation defects of both mutants.

From a structural point of view, an interesting question is how the two mutations, which affect channel gating by ATP in very different ways, nevertheless cause such similar defects in channel activation by PKA. Indeed, although mutation ΔF508 targets the NBD1–TMD interface, whereas G551D targets the NBD1–NBD2 interface (Fig. 1*B*), many of their structural consequences are shared. First, both mutations destabilize the NBD dimer interface (Jih et al., 2011; Kopeikin et al., 2014; Lin et al., 2014; Wang et al., 2022), which can be stabilized by P-ATP (Jih et al., 2011; Wang et al., 2022). Second, the 'ball-and-socket'-type NBD1–TMD interaction is also destabilized not only by the ΔF508 mutation, which physically targets the 'socket' (Fiedorczuk & Chen, 2022), but also by mutation G551D (Wang et al.,

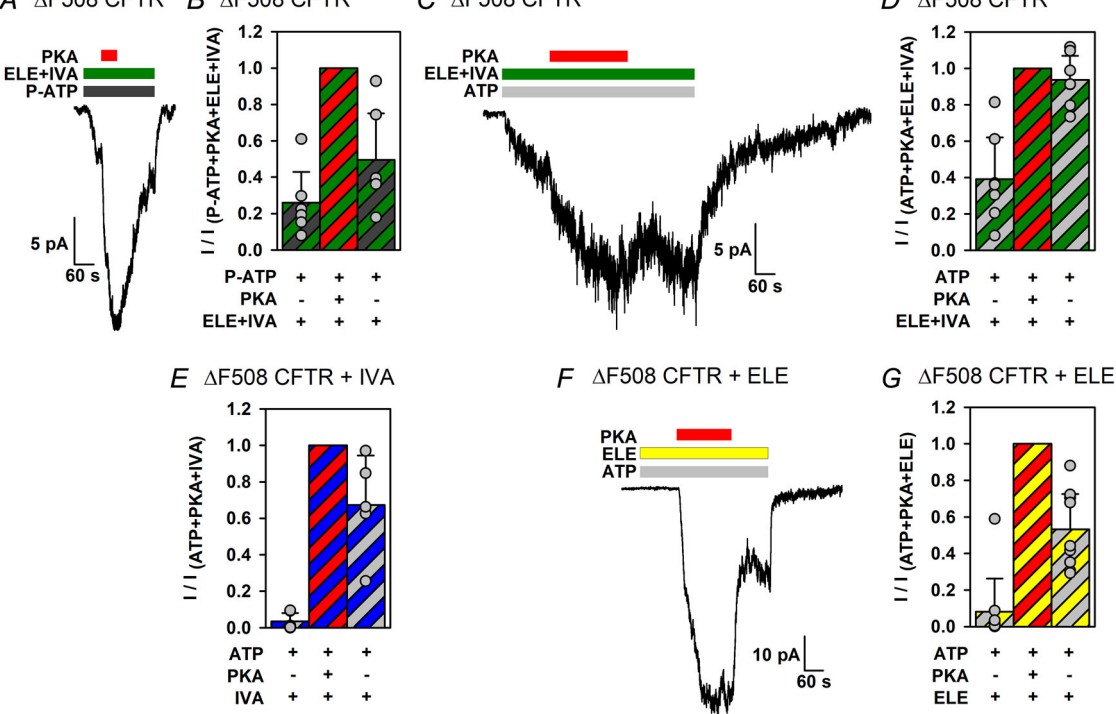

**Figure 8. Effects of potentiator drugs on non-catalytic and catalytic activation of ΔF508 CFTR**
*A*, macroscopic inside-out patch current of unphosphorylated ΔF508 CFTR channels activated by exposure to 10 µM P-ATP (dark grey bar) in the presence of 1 µM elexacaftor (ELE) + 10 nM ivacaftor (IVA) (green bar), and response to application of 300 nM PKA (red bar). *B*, mean steady-state currents in the three consecutive segments of the experimental protocol shown in *A*, normalized to the mean current observed during PKA exposure. Bars show the mean ± SD, $n = 6$. *C* and *F*, macroscopic inside-out patch currents of ΔF508 CFTR channels in the presence of 2 mM ATP (light grey bars) and either 1 µM elexacaftor + 10 nM ivacaftor (*C*, green bar) or 1 µM elexacaftor alone (*F*, yellow bar). Steady-state currents were sampled before, during and after full channel activation by a 3–4 min superfusion with 300 nM PKA (red bars). *D*, *E* and *G*, fractional steady-state currents of ΔF508 CFTR channels in 2 mM ATP and indicated drug(s) before PKA exposure (first bars), in the presence of 300 nM PKA (second bars) and following PKA removal (third bars). Data in *E* are replotted from Fig. 2*K* of Mihályi et al. (2024). Bars show the mean ± SD, $n = 7$ (*D*), $n = 5$ or 6 (*E*) and $n = 9$ (*F*). The ratio of the current amplitude following PKA removal to that in the presence of PKA (cf. *B*, *D*, *E* and *G*, third bar) was significantly smaller than unity in *B* ($P = 2.37 \times 10^{-3}$), *E* ($P = 0.0271$) and *G* ($P = 4.32 \times 10^{-5}$), but not in *D* ($P = 0.125$) (Student's unpaired, one-tailed *t* test).

2022). Thus, both mutations result in a loosening of NBD1–TMD structural coupling. Interestingly, P-ATP not only stabilizes the NBD dimer interface, but also the NBD1–TMD interface for G551D CFTR (Wang et al., 2022). These findings suggest that non-catalytic activation by PKA requires either a stable NBD dimer and/or a stable NBD1–TMD interface. In the structure of the non-catalytically activated CFTR–PKA complex (PDBID: 9dw9), an R-domain loop (residues 806–833) is docked onto the outer surface of the NBD1–TMD interface (Fig. 1B, red surface). One possible structural explanation for our findings is that the conformational change that leads to reversible channel stimulation by PKA might involve docking of that R-domain loop. If so, the mutations might impair non-catalytic stimulation by destabilizing the NBD1–TMD interface, which forms the docking site for the upper arm (residues 819–833) of that loop, and P-ATP might enhance non-catalytic stimulation for the mutants by stabilizing that interface.

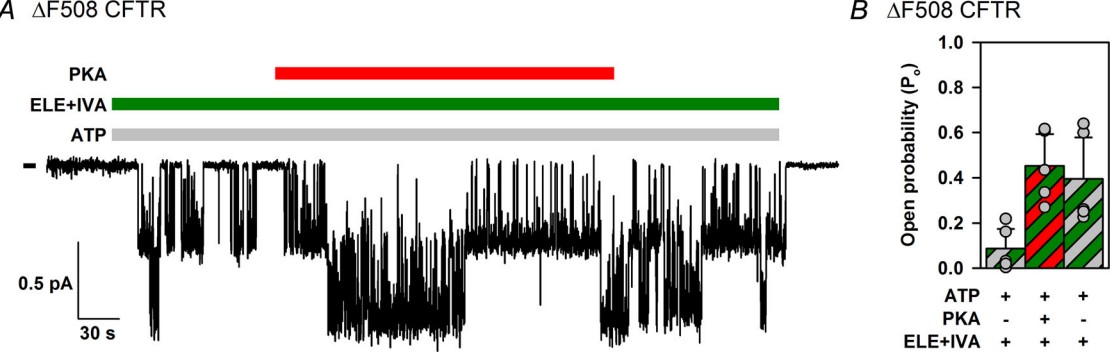

**Figure 9. In the presence of ivacaftor + elexacaftor, open probability of phosphorylated ΔF508 CFTR does not approach unity**
*A*, microscopic inside-out patch current from two active ΔF508 CFTR channels in the presence of 2 mM ATP (light grey bar) and 1 μM elexacaftor (ELE) + 10 nM ivacaftor (IVA) (green bar). Steady-state gating was sampled before, during and after full channel activation by a 4 min superfusion with 300 nM PKA (red bar). The recording temperature was 25°C. The dash before the current trace marks the zero-current level. *B*, open probabilities ($P_o$) of ΔF508 CFTR channels in 2 mM ATP and 1 μM elexacaftor + 10 nM ivacaftor before PKA exposure (first bar), in the presence of 300 nM PKA (second bar) and following PKA removal (third bar). Bars show the mean ± SD, $n = 5$. The $P_o$ following PKA removal was not significantly smaller than that in the presence of PKA ($P = 0.297$, Student's paired, one-tailed *t* test).

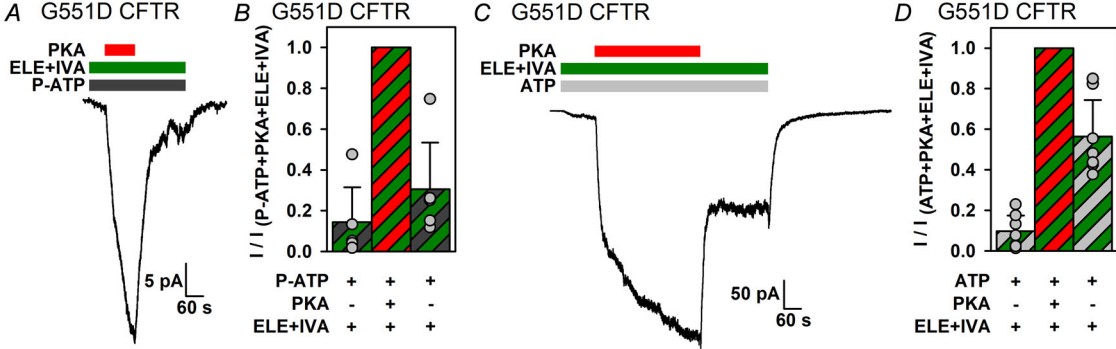

**Figure 10. Effects of potentiator drugs on non-catalytic and catalytic activation of G551D CFTR**
*A*, macroscopic inside-out patch current of unphosphorylated G551D CFTR channels activated by exposure to 10 μM P-ATP (dark grey bar) in the presence of 1 μM elexacaftor (ELE) + 10 nM ivacaftor (IVA) (green bar), and response to application of 300 nM PKA (red bar). *B*, mean steady-state currents in the three consecutive segments of the experimental protocol shown in *A*, normalized to the mean current observed during PKA exposure. Bars show the mean ± SD, $n = 5$. *C*, macroscopic inside-out patch current of G551D CFTR channels in the presence of 2 mM ATP (light grey bar) and 1 μM elexacaftor + 10 nM ivacaftor (green bar). Steady-state currents were sampled before, during and after full channel activation by a 4 min superfusion with 300 nM PKA (red bar). *D*, mean steady-state currents in the three consecutive segments of the experimental protocol shown in *C*, normalized to the mean current observed during PKA exposure. Bars show the mean ± SD, $n = 7$. The ratio of the current amplitude following PKA removal to that in the presence of PKA (*B* and *D*, third bar) was significantly smaller than unity both in *B* ($P = 1.21 \times 10^{-3}$) and in *D* ($P = 3.36 \times 10^{-4}$) (Student's unpaired, one-tailed *t* test).

The intact affinities of both mutants for PKA (Fig. 7) seem to contrast with the earlier observation of a large rightward shift in the dose–response curve for current activation by PKA in inside-out patches for the G551D mutant relative to WT CFTR (Wang et al., 2020). That discrepancy calls for a reinterpretation of what a PKA dose–response curve means. Early studies (Hwang et al., 1994; Szellas & Nagel, 2003) postulated the presence of endogenous phosphatases in the patch membrane. Hence, PKA dose–response curves (Csanády et al., 2005; Wang et al., 2020) were conventionally interpreted to report the effect of gradually increasing steady-state stoichiometries of R-domain phosphorylation. That view was overthrown by the recent demonstration that the rapid partial current decline following PKA removal reflects loss of non-catalytic channel stimulation by bound PKA, rather than the action of phosphatases. Indeed, phosphatases are unlikely to be present in excised patches, because full CFTR channel phosphorylation can be reached even with low PKA concentrations if the exposure time is long enough (Mihályi et al., 2020). In contrast, non-catalytic activation requires higher free PKA concentrations (Mihályi et al., 2020). Because, for G551D CFTR, the amplitude of the catalytically activated current component is very small (Fig. 4*F*), substantial currents are evoked only by PKA concentrations that are sufficiently high to cause non-catalytic activation. Of note, in the earlier study (Wang et al., 2020), stimulation of G551D CFTR channels by PKA at concentrations up to ∼2.4 µM (Sigma P2645, 858 U/ml) might have been influenced by inorganic phosphate ($P_i$) present in that preparation (∼40 mM $P_i$ in 2.4 µM PKA), which strongly stimulates at least WT CFTR (Mihályi et al., 2020).

In an era when most CF patients receive ETI treatment, a question of interest is to what extent the two potentiators of that drug combination (elexacaftor and ivacator) affect non-catalytic *vs.* catalytic channel activation by PKA. We show here that, for both mutants, elexacaftor + ivacaftor boosts the catalytically activated (irreversible) component more efficiently than the non-catalytically activated (reversible) component, such that the fractional contribution of the latter to the total current is diminished in the presence of the drugs (compare Fig. 8*C* and *D* with Fig. 4*C* and *G*, and Fig. 10*C* and *D* with Fig. 4*F* and *H*). Indeed, for phosphorylated ΔF508 CFTR exposed to both drugs, non-catalytic stimulation is barely detectable (Fig. 8*C* and *D*). One potential explanation might be that, in the presence of elexacaftor + ivacaftor, the $P_o$ of phosphorylated ΔF508 CFTR channels approaches ∼1 even in the absence of PKA. However, kinetic analysis of single-channel recordings suggests that the $P_o$ in such conditions is only ∼0.5 (Fig. 9), consistent with a recent study in mammalian cells on ΔF508 CFTR rescued by chronic ETI treatment and acutely potentiated with elexacaftor + ivacaftor

in the presence of ATP + PKA (Rodrat et al., 2026). Alternatively, drug binding might interfere with docking of PKA at the NBD1–TMD interface (Fig. 1*B*), required for non-catalytic stimulation. However, that possibility is also unlikely, because non-catalytic stimulation is clearly present for unphosphorylated ΔF508 channels in the presence of elexacaftor + ivacaftor (Fig. 8*A* and *B*), or for phosphorylated ΔF508 channels in the presence of either ivacaftor (Fig. 8*E*) or elexacaftor (Fig. 8*F* and *G*) alone. Moreover, neither drug binding site overlaps with that of PKA (Fig. 1*B*). Whatever the underlying reason, these findings suggest room for further modulator improvement to achieve drug combinations that boost not only catalytic but also non-catalytic activation of phosphorylated ΔF508 CFTR channels by PKA, as demonstrated for P-ATP. Furthermore, a drug combination that preserves strict PKA dependence of ΔF508 CFTR channel activity might also represent a step forwards, in comparison to elexacaftor + ivacaftor, which evokes substantial PKA-independent activity for that mutant (Fig. 8*D*, first bar).

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

## Additional information

### Data availability statement

All data that support the findings of this study are available within the article.

### Competing interests

The authors declare no conflicts of interest.

### Author contributions

O.Z., M.A.S. and L.C. designed research. O.Z. and M.A.S. performed research. O.Z., M.A.S. and L.C. analysed data. O.Z. and L.C. wrote the paper.

### Funding

Supported by National Research, Development and Innovation Office grants KKP 144199 to L.C. and TKP2021-EGA-24 to Semmelweis University, and by Cystic Fibrosis Foundation Research grant CSANAD21G0 to L.C.

### Acknowledgements

We thank Drs Andras Szollosi and Iordan Iordanov for providing the purified PKA catalytic subunit.

### Keywords

cystic fibrosis transmembrane conductance regulator, cystic fibrosis, elexacaftor, ivacaftor, protein kinase A

## Supporting information

Additional supporting information can be found online in the Supporting Information section at the end of the HTML view of the article. Supporting information files available:

**Peer Review History**

