## [Peer Review History · The Journal of Physiology]

Altered functional interactions between CFTR disease mutants DeltaF508 and G551D and the protein kinase A catalytic subunit

Olivér Závoti, Márton A Simon, and László Csanády

DOI: 10.1113/JP290798

Corresponding author(s): László Csanády (csanady.laszlo@semmelweis.hu)

Review Timeline:

Submission Date:	22-Dec-2025
Editorial Decision:	27-Jan-2026
Revision Received:	09-Feb-2026
Editorial Decision:	20-Feb-2026
Revision Received:	20-Feb-2026
Accepted:	25-Feb-2026

Senior Editor: *Peying Fong*

Reviewing Editor: *Péter Hegyi*

Transaction Report:

Dear Dr Csanády,

Re: JP-RP-2025-290798 "Altered functional interactions between CFTR disease mutants DeltaF508 and G551D and the protein kinase A catalytic subunit" by Olivér Závoti, Márton A Simon, and László Csanády

Thank you for submitting your manuscript to The Journal of Physiology. It has been assessed by a Reviewing Editor and by 2 expert referees and we are pleased to tell you that it is acceptable for publication following satisfactory revision.

REVISION CHECKLIST:

We look forward to receiving your revised submission.

Yours sincerely,

Peying Fong
Senior Editor
The Journal of Physiology

REQUIRED ITEMS

1) - Author photo and profile. First or joint first authors are asked to provide a short biography (no more than 100 words for one author or 150 words in total for joint first authors) and a portrait photograph. These should be uploaded and clearly labelled together in a Word document with the revised version of the manuscript. See Information for Authors for further details.

2) - You must start the Methods section with a paragraph headed Ethical approval (https://jp.msubmit.net/cgi-bin/main.plex?form_type=display_requirements#methods).

Research must comply with The Journal's policies regarding animal experiments (<https://physoc.onlinelibrary.wiley.com/hub/animal-experiments>) and adherence to these policies must be stated in the manuscript.

Authors should confirm in their Methods section that their experiments were carried out according to the guidelines laid down by their institution's animal welfare committee, including an ethics approval reference number. The Methods section must contain a statement about access to food, water and housing, details of the anaesthetic regime: anaesthetic used, dose and route of administration, and method of killing the experimental animals.

3) - Please upload separate high-quality figure files via the submission form.

4) - Please ensure that the Article File you upload is a Word file.

5) - Your paper contains Supporting Information of a type that we no longer publish, including supplementary tables and figures. Any information essential to an understanding of the paper must be included as part of the main manuscript and figures. The only Supporting Information that we publish are video and audio, 3D structures, program codes and large data files. Your revised paper will be returned to you if it does not adhere to our Supporting Information Guidelines.

6) - Papers must comply with the Statistics Policy: https://jp.msubmit.net/cgi-bin/main.plex?form_type=display_requirements#statistics.

In summary:

- If n {less than or equal to} 30, all data points must be plotted in the figure in a way that reveals their range and distribution. A bar graph with data points overlaid, a box and whisker plot or a violin plot (preferably with data points included) are acceptable formats.

- If $n > 30$, then the entire raw dataset must be made available either as supporting information, or hosted on a not-for-profit repository, e.g. FigShare, with access details provided in the manuscript.
- 'n' clearly defined (e.g. x cells from y slices in z animals) in the Methods. Authors should be mindful of pseudoreplication.
- All relevant 'n' values must be clearly stated in the main text, figures and tables.
- The most appropriate summary statistic (e.g. mean or median and standard deviation) must be used. Standard Error of the Mean (SEM) alone is not permitted.
- Exact p values must be stated. Authors must not use 'greater than' or 'less than'. Exact p values must be stated to three significant figures even when 'no statistical significance' is claimed.

7) - Please include an Abstract Figure file and an Abstract Figure legend. An appropriate figure legend, which should not exceed 150 words in length, should be included in the main manuscript file. The Abstract Figure is a piece of artwork designed to give readers an immediate understanding of the research and should summarise the main conclusions. If possible, the image should be easily 'readable' from left to right or top to bottom. It should show the physiological relevance of the manuscript so readers can assess the importance and content of its findings. Abstract Figures should not merely recapitulate other figures in the manuscript. Please try to keep the diagram as simple as possible and without superfluous information that may distract from the main conclusion(s). Abstract Figures must be provided by authors no later than the revised manuscript stage and should be uploaded as a separate file during online submission labelled as File Type 'Abstract Figure'. Please also ensure that you include the figure legend in the main article file. All Abstract Figures should be created using BioRender. Authors should use The Journal's premium BioRender account to export high-resolution images. Details on how to use and access the premium account are included as part of this email.

8) - Please include a full title page as part of your main article (Word) file, which should contain the following: title, authors, affiliations, corresponding author name and contact details, keywords, and running title.

9) - Please ensure that all figures and tables have a title and legend, and that they have been cited within the main article text.

EDITOR COMMENTS

Reviewing Editor:

Both reviewers find your manuscript to be carefully designed, clearly written, and of high potential impact in the field of CFTR regulation and cystic fibrosis research. The study is considered suitable for publication after minor revisions. No additional experiments are requested.

To further strengthen the manuscript, please address the following points:

- One reviewer noted potential ambiguity in the use of the terms "reversible" and "irreversible" PKA-dependent activation. Please clarify in the Discussion that "irreversible" refers to activation persisting over the experimental time course, rather than absolute irreversibility, and that dephosphorylation would be expected to reverse this effect over longer timescales.
- You suggest that slowed activation by PKA phosphorylation in the mutants may reflect delayed attainment of full phosphorylation. As the extent of phosphorylation was not directly measured, please expand the Discussion to acknowledge alternative, non-mutually exclusive explanations, such as mutation-induced alterations in phosphorylation site accessibility or in conformational transitions downstream of phosphorylation.
- While the experimental design and dataset are strong, the statistical treatment could be expanded. Please clarify in the Methods and Results which comparisons were subjected to statistical testing, and consider applying paired statistical analyses (e.g. paired t-tests) where measurements were obtained from the same patch. This is particularly relevant for macroscopic current data and would further strengthen the robustness of the conclusions.
- Direct comparison of PKA effects across conditions would be facilitated by using identical scales for representative traces where feasible (e.g. Figures 2, 7, and 8).

In addition, please consider whether the $\Delta F508$ trace shown in Figure 5 is the most representative example of the

irreversible component following extensive phosphorylation.

- Please address the minor wording suggestion on page 16 regarding the description of P-ATP effects on the irreversible component, as noted by the reviewer. The reviewers' more speculative mechanistic considerations are thought-provoking but optional; inclusion is at your discretion and not required for acceptance.

Senior Editor:

Initial review of your manuscript, "Altered functional interactions between CFTR disease mutants DeltaF508 and G551D and the protein kinase A catalytic subunit" is now complete. It has been scrutinized by two Expert Referees and a Reviewing Editor. From the attached reports, you will read that, all found your study to be potentially impactful. Furthermore, they concur that no further experiments are necessary for this manuscript, although Referee 2 does enthusiastically offer some final thoughts on what lines of inquiry your study might further catalyze. Taken together, understanding alterations in molecular mechanisms germane to protein kinase A regulation of the two common, disease-causing CFTR mutants, DeltaF508 and G551D, is important. Ultimately, this knowledge can prove foundational toward better leveraging available drug therapies.

Nonetheless, both Referees and the Reviewing Editor also offer several suggestions for refining your manuscript. As you will read, these primarily include clarification of terminology (Referee 1's final comment; highlighted in RE summary) and optimization of figural data presentation, as well as incorporating paired statistical comparisons (Referee 2; see also RE summary).

To the point of statistics, please do ensure that presentation is fully compliant with The Journal of Physiology's published Statistic Policies. I note that the supplemental figures are of the type that will require incorporation into the body of the manuscript proper; please ensure that this change is also incorporated in your revised manuscript.

Thank you for favoring The Journal of Physiology with this interesting and important study. We look forward to receiving your revised manuscript.

Statistical methods do not fully comply with The Journal of Physiology's Statistics Policy. There is a need to better define n ; presently this is the number of independent experiments, but precisely what is meant by what an independent experiment should be better defined. Is it correct to surmise this means individual patches?

Please also note that exact p values, to three significant figures, do not appear to be given in this initial submission.

REFEREE COMMENTS

Referee #1:

This group has previously demonstrated that CFTR is activated not only by phosphorylation of multiple residues in the channel but also by direct binding of Protein Kinase A (PKA) to the protein. In the present paper by Zavoti and colleagues, PKA modulation of common mutations in CFTR are investigated. Further, they determine how the different modes of PKA activation of these variants are altered by clinically relevant therapeutic agents. Specifically, they show that both $\Delta F508$ and G551D are activated directly by PKA binding and this effect in contrast to WT CFTR was greatly enhanced if CFTR is previously phosphorylated. Nevertheless, the effect of phosphorylation of the channel to increase activity was compromised to a greater extent than direct PKA binding in both mutants. In addition, they demonstrate that the direct effect of PKA binding is lost in the presence of a "CFTR potentiator" combination but not by individual compounds.

The study is a clear progression of careful experiments and the paper is well written and logically presented. I have a few relatively minor comments.

I am a little confused by the terminology used to describe the effects on CFTR activation by direct PKA binding as "reversible" and PKA-mediated phosphorylation as "irreversible". Since, I presume that if the channel is dephosphorylated the activation is in fact reversed, does irreversible in this context refer to "not reversed in the time period of the experiment" vs. the readily reversible effect on activity following dissociation of PKA. Some clarification in the discussion might help the reader.

The experiments show that activation by PKA phosphorylation is slowed in the mutants and the Authors suggest that this is because the mutant is slower to become fully phosphorylated. Since the extent of phosphorylation was not directly measured can they discriminate between the mutant altering the kinetics of phosphorylation or particular sites of phosphorylation per se or that the mutants disrupt conformational changes induced by phosphorylation.

Direct comparison of the effects of PKA on the mutants would be helped by plotting representative traces with the same scales (eg fig 2, fig 7, 8).

Referee #2:

This is an interesting paper, investigating how PKA activates two cystic fibrosis (CF)-causing CFTR variants.

Impact

The authors include little in terms of broader speculative considerations on how their results could affect the scientific understanding in this context. This is reasonable, given the early stage. However, I think these studies address a significant knowledge gap, extending recent advances in our understanding of the molecular mechanisms of CFTR regulation, to the effects of disease-causing mutations on those mechanisms. This considerably increases the translational potential.

Insight into physiological mechanisms

These studies could form the foundation of future programmes attempting at improving efficacy of therapeutic drugs. We are far from that, at the moment, but understanding basic molecular underpinning of PKA regulation of CFTR, and how this is affected by mutations carried by the majority of CF patients, is a first, essential step.

Originality of the research

Elegant experimental procedures are used to discriminate reversible and irreversible PKA-dependent activation of CFTR. These techniques have been pioneered by the lab of Prof Csanády, based on advances made in his and other labs over the past decades.

Study design and robustness of the experimental data; Validity of conclusions

Experiments are well designed and include a high number of experimental repeats. However, the statistical analysis is rather limited. Methods indicate that t-tests were used, but I could only find one mentioned. While the authors indicate that some of the single-channel kinetic parameter estimates are not identified with precision because of the inherent difficulties of the experiment, I think statistical analyses could be used effectively for many of the macroscopic current experiments. Simple paired t-test, pairing measurements obtained from the same patch, possibly with some multiple comparison correction, would probably be enough. This could, in my opinion, strengthen the conclusions considerably.

Minor points

Page 16: "For both mutants, earlier studies had reported stimulation of the irreversible component, i.e. the P_o of phosphorylated channels gating in the absence of PKA, by P-ATP". Possibly better:

"For both mutants, earlier studies had reported stimulation of the irreversible component by P-ATP, i.e. an increased P_o of phosphorylated channels gating in the absence of PKA"

Fig 5: The $\Delta F508$ CFTR trace displayed does not show a clear increase in the irreversible current component, following extensive phosphorylation. Could a more representative trace be used?

Speculative thoughts (can be ignored by editors and authors alike)

I find it extremely interesting that, for phosphorylated $\Delta F508$ CFTR, once potentiated by ellexacaftor and ivacaftor, the additional, reversible PKA effect is largely lost. Might it be possible that R-domain phosphorylation, docking of the R domain loop (possibly reflecting the reversible PKA stimulation), ellexacaftor and ivacaftor binding, and possibly P-ATP all contribute to stabilization of an ICL4/ICL1/NBD1/NBD2 complex which is permissive for an open permeation pathway? There is

evidence that the $\Delta F508$ mutation renders the NBD1 structure much more dynamically linked to the rest of the protein (K. Fiedorczuk and J. Chen 2022, DOI: 10.1126/science.ade2216; D. Scholl et al. 202, DOI: 10.1038/s41589-021-00844-0). As the authors note, in some conditions at least, G551D also seems to also make the NBD1/TMD joint more dynamic (C. Wang et al., 2022, DOI: 10.1101/2022.10.10.510913). It seems plausible to me that all the above factors could synergistically stabilize the ICL4/ICL1/NBD1/NBD2 complex, but that this stabilization could reach a ceiling, beyond which further stabilization would be negligible. For instance, when $\Delta F508$ CFTR is phosphorylated, and elxacaftor- and ivacaftor-bound, additional PKA binding might provide no further stability. Thus, PKA's reversible docking would not be blocked, but rather made redundant. The authors mention a possible ceiling at P_o reaching ~ 1 . But the state including the multidomain complex might be in dynamic equilibrium with the open-permeation pathway state, with a further transition required for channel opening. This ceiling could sit at $P_o=0.4-0.5$, as the authors observe (figure S2).

END OF COMMENTS

EDITOR COMMENTS

Reviewing Editor:

Both reviewers find your manuscript to be carefully designed, clearly written, and of high potential impact in the field of CFTR regulation and cystic fibrosis research. The study is considered suitable for publication after minor revisions. No additional experiments are requested.

To further strengthen the manuscript, please address the following points:

- One reviewer noted potential ambiguity in the use of the terms "reversible" and "irreversible" PKA-dependent activation. Please clarify in the Discussion that "irreversible" refers to activation persisting over the experimental time course, rather than absolute irreversibility, and that dephosphorylation would be expected to reverse this effect over longer timescales.

Done, see responses to Referee #1.

- You suggest that slowed activation by PKA phosphorylation in the mutants may reflect delayed attainment of full phosphorylation. As the extent of phosphorylation was not directly measured, please expand the Discussion to acknowledge alternative, non-mutually exclusive explanations, such as mutation-induced alterations in phosphorylation site accessibility or in conformational transitions downstream of phosphorylation.

Done, see responses to Referee #1.

- While the experimental design and dataset are strong, the statistical treatment could be expanded. Please clarify in the Methods and Results which comparisons were subjected to statistical testing, and consider applying paired statistical analyses (e.g. paired t-tests) where measurements were obtained from the same patch. This is particularly relevant for macroscopic current data and would further strengthen the robustness of the conclusions.

We have now performed statistical tests for the data sets shown in Figures 2, 4, 5, 6, 8, 9, and 10. Exact p values are provided in the figure legends, as specified in Methods.

- Direct comparison of PKA effects across conditions would be facilitated by using identical scales for representative traces where feasible (e.g. Figures 2, 7, and 8).

Done (revised Figures 2, 8, and 10).

In addition, please consider whether the $\Delta F508$ trace shown in Figure 5 is the most representative example of the irreversible component following extensive phosphorylation.

We have replaced the current trace with a more representative record (revised Figure 6).

- Please address the minor wording suggestion on page 16 regarding the description of P-ATP effects on the irreversible component, as noted by the reviewer. The reviewers' more speculative mechanistic considerations are thought-provoking but optional; inclusion is at your discretion and not required for acceptance.

Corrected, thank you.

Senior Editor:

Initial review of your manuscript, "Altered functional interactions between CFTR disease mutants DeltaF508 and G551D and the protein kinase A catalytic subunit" is now complete. It has been scrutinized by two Expert Referees and a Reviewing Editor. From the attached reports, you will read that, all found your study to be potentially impactful. Furthermore, they concur that no further experiments are necessary for this manuscript, although Referee 2 does enthusiastically offer some final thoughts on what lines of inquiry your study might further catalyze. Taken together, understanding alterations in molecular mechanisms germane to protein kinase A regulation of the two common, disease-causing CFTR mutants, DeltaF508 and G551D, is important. Ultimately, this knowledge can prove foundational toward better leveraging available drug therapies.

Nonetheless, both Referees and the Reviewing Editor also offer several suggestions for refining your manuscript. As you will read, these primarily include clarification of terminology (Referee 1's final comment; highlighted in RE summary) and optimization of figural data presentation, as well as incorporating paired statistical comparisons (Referee 2; see also RE summary).

All done, according to the suggestions of Referee 2 and the Reviewing Editor.

To the point of statistics, please do ensure that presentation is fully compliant with The Journal of Physiology's published Statistic Policies. I note that the supplemental figures are of the type that will require incorporation into the body of the manuscript proper; please ensure that this change is also incorporated in your revised manuscript.

We have moved both supplementary figures among the main-text figures. Fig. S1 has become Fig. 5 and is now discussed in the main text on P13 (2nd par). Fig. S2 is now listed as Fig. 9, and discussed on P15 (3rd par). The numbering of original figures 5-8 has been adjusted accordingly.

Statistical methods do not fully comply with The Journal of Physiology's Statistics Policy. There is a need to better define n; presently this is the number of independent experiments, but precisely what is meant by what an independent experiment should be better defined. Is it correct to surmise this means individual patches?

Yes, independent experiments mean individual patches, this is now specified in Methods (P9).

Please also note that exact p values, to three significant figures, do not appear to be given in this initial submission.

These are now provided in the Figure legends.

REFEREE COMMENTS

Referee #1:

This group has previously demonstrated that CFTR is activated not only by phosphorylation of multiple residues in the channel but also by direct binding of Protein Kinase A (PKA) to the protein. In the present paper by Zavoti and colleagues, PKA modulation of common mutations in CFTR are investigated. Further, they determine how the different modes of PKA activation of these variants are altered by clinically relevant therapeutic agents. Specifically, they show that both $\Delta F508$ and G551D are activated directly by PKA binding and this effect in contrast to WT

CFTR was greatly enhanced if CFTR is previously phosphorylated. Nevertheless, the effect of phosphorylation of the channel to increase activity was compromised to a greater extent than direct PKA binding in both mutants. In addition, they demonstrate that the direct effect of PKA binding is lost in the presence of a "CFTR potentiator" combination but not by individual compounds.

The study is a clear progression of careful experiments and the paper is well written and logically presented. I have a few relatively minor comments.

We thank the Reviewer for the positive evaluation and insightful comments regarding the clarity of presentation, which we address below.

I am a little confused by the terminology used to describe the effects on CFTR activation by direct PKA binding as "reversible" and PKA-mediated phosphorylation as "irreversible". Since, I presume that if the channel is dephosphorylated the activation is in fact reversed, does irreversible in this context refer to "not reversed in the time period of the experiment" vs. the readily reversible effect on activity following dissociation of PKA. Some clarification in the discussion might help the reader.

We thank the Reviewer for pointing out this ambiguity. The "irreversible" component typically survives for the entire duration of our experiments, because in our inside-out patches there are no membrane-attached phosphatases that would dephosphorylate CFTR (see Mihályi et al., 2020). We now explain this on P14 (end of par 1). In addition, to avoid confusion, we have replaced "reversible activation" and "irreversible activation" throughout the text by "noncatalytic activation" and "catalytic activation", respectively (as also used in Mihályi et al., 2024, and Fiedorczuk et al., 2024). (But we have kept the qualifiers "reversible" and "irreversible" when referring to current components, for which "noncatalytic" and "catalytic" would be inappropriate.)

The experiments show that activation by PKA phosphorylation is slowed in the mutants and the Authors suggest that this is because the mutant is slower to become fully phosphorylated. Since the extent of phosphorylation was not directly measured can they discriminate between the mutant altering the kinetics of phosphorylation or particular sites of phosphorylation per se or that the mutants disrupt conformational changes induced by phosphorylation.

We did not explicitly state that phosphorylation *per se* is slowed for the mutants. We simply reported that the irreversible (catalytically activated) current component develops slower for $\Delta F508$ CFTR. It is true though, that the above statement is implicit in the model in Fig. 1A: in the framework of that model catalytic activation is represented by step 3→4, which reflects phosphorylation. We thus agree with the Reviewer that more discussion on that topic is warranted. We now explicitly discuss both possible explanations, but point out that the alternative interpretation seems less likely (P17, last par):

"Based on the scheme (Fig. 1A), in which catalytic activation (step 3→4) reflects R-domain phosphorylation, the slowed development of the irreversible current component for $\Delta F508$ CFTR would suggest that the mutant is phosphorylated slower (cf., (Pasyk et al., 2009)). It is formally possible that catalytic activation is not rate limited by phosphorylation per se, but by the conformational changes induced by phosphorylation, and that the mutation slows those latter steps. However, if so, then following a short (1-minute) PKA application, which only submaximally activates the irreversible current component, that component (which survives sudden PKA removal) should continue to rise revealing slow completion of those conformational steps. Such behaviour was observed neither for WT (Mihályi et al., 2024) nor for $\Delta F508$ CFTR (Fig. 5A), suggesting that activation of the catalytic current component is indeed rate limited by phosphorylation for both WT and the mutant."

Direct comparison of the effects of PKA on the mutants would be helped by plotting representative traces with the same scales (eg fig 2, fig 7, 8).

Thank you, we now show all panels at the same time scale in all three indicated figures (revised Figures 2, 8, and 10). (Current scales cannot be compared as each patch contains different numbers of channels.)

Referee #2:

This is an interesting paper, investigating how PKA activates two cystic fibrosis (CF)-causing CFTR variants.

Impact

The authors include little in terms of broader speculative considerations on how their results could affect the scientific understanding in this context. This is reasonable, given the early stage. However, I think these studies address a significant knowledge gap, extending recent advances in our understanding of the molecular mechanisms of CFTR regulation, to the effects of disease-causing mutations on those mechanisms. This considerably increases the translational potential.

Insight into physiological mechanisms

These studies could form the foundation of future programmes attempting at improving efficacy of therapeutic drugs. We are far from that, at the moment, but understanding basic molecular underpinning of PKA regulation of CFTR, and how this is affected by mutations carried by the majority of CF patients, is a first, essential step.

Originality of the research

Elegant experimental procedures are used to discriminate reversible and irreversible PKA-dependent activation of CFTR. These techniques have been pioneered by the lab of Prof Csanády, based on advances made in his and other labs over the past decades.

We thank the Reviewer for a positive evaluation, and for several constructive comments.

Study design and robustness of the experimental data; Validity of conclusions

Experiments are well designed and include a high number of experimental repeats. However, the statistical analysis is rather limited. Methods indicate that t-tests were used, but I could only find one mentioned. While the authors indicate that some of the single-channel kinetic parameter estimates are not identified with precision because of the inherent difficulties of the experiment, I think statistical analyses could be used effectively for many of the macroscopic current experiments. Simple paired t-test, pairing measurements obtained from the same patch, possibly with some multiple comparison correction, would probably be enough. This could, in my opinion, strengthen the conclusions considerably.

Thank you for this suggestion. To support our claims, we have now performed statistical tests for the data sets shown in Figures 2, 4, 5, 6, 8, 9, and 10. Exact p values are provided in the figure legends.

Minor points

Page 16: "For both mutants, earlier studies had reported stimulation of the irreversible component, i.e. the P_o of phosphorylated channels gating in the absence of PKA, by P-ATP". Possibly better: "For both mutants, earlier studies had reported stimulation of the irreversible component by P-ATP, i.e. an increased P_o of phosphorylated channels gating in the absence of PKA"

Corrected, thank you.

Fig 5: The $\Delta F508$ CFTR trace displayed does not show a clear increase in the irreversible current component, following extensive phosphorylation. Could a more representative trace be used?

We have replaced the current trace with a more representative record.

Speculative thoughts (can be ignored by editors and authors alike)

I find it extremely interesting that, for phosphorylated $\Delta F508$ CFTR, once potentiated by elxacaftor and ivacaftor, the additional, reversible PKA effect is largely lost. Might it be possible that R-domain phosphorylation, docking of the R domain loop (possibly reflecting the reversible PKA stimulation), elxacaftor and ivacaftor binding, and possibly P-ATP all contribute to stabilization of an ICL4/ICL1/NBD1/NBD2 complex which is permissive for an open permeation pathway? There is evidence that the $\Delta F508$ mutation renders the NBD1 structure much more dynamically linked to the rest of the protein (K. Fiedorczuk and J. Chen 2022, DOI: 10.1126/science.ade2216; D. Scholl et al. 202, DOI: 10.1038/s41589-021-00844-0). As the authors note, in some conditions at least, G551D also seems to also make the NBD1/TMD joint more dynamic (C. Wang et al., 2022, DOI: 10.1101/2022.10.10.510913). It seems plausible to me that all the above factors could synergistically stabilize the ICL4/ICL1/NBD1/NBD2 complex, but that this stabilization could reach a ceiling, beyond which further stabilization would be negligible. For instance, when $\Delta F508$ CFTR is phosphorylated, and elxacaftor- and ivacaftor-bound, additional PKA binding might provide no further stability. Thus, PKA's reversible docking would not be blocked, but rather made redundant. The authors mention a possible ceiling at P_o reaching ~ 1 . But the state including the multidomain complex might be in dynamic equilibrium with the open-permeation pathway state, with a further transition required for channel opening. This ceiling could sit at $P_o=0.4-0.5$, as the authors observe (figure S2).

We thank the Reviewer for these interesting thoughts. We have finally decided not to include them in the Discussion, as there is currently no experimental evidence to support them.

Dear Dr Csanády,

Re: JP-RP-2026-290798R1 "Altered functional interactions between CFTR disease mutants DeltaF508 and G551D and the protein kinase A catalytic subunit" by Olivér Závoti, Márton A Simon, and László Csanády

Thank you for submitting your manuscript to The Journal of Physiology. It has been assessed by a Reviewing Editor and by 2 expert referees and we are pleased to tell you that it is acceptable for publication following satisfactory revision.

REVISION CHECKLIST:

We look forward to receiving your revised submission.

Yours sincerely,

Peying Fong
Senior Editor
The Journal of Physiology

REQUIRED ITEMS

EDITOR COMMENTS

Reviewing Editor:

Comments to the Author (Required):

Please revise the manuscript by implementing the minor textual edits suggested by Referee #2 (replacement of "quantitated" with "analyzed" on page 13 and rephrasing of the sentence on page 18 for clarity) before final acceptance.

Senior Editor:

The reviews of your carefully revised manuscript, "Altered functional interactions between CFTR disease mutants DeltaF508 and G551D and the protein kinase A catalytic subunit" are now available and are attached herewith. Both original Referees concur on its potential for contributing impactfully to the collective knowledge of CFTR regulation, as do both I and the Reviewing Editor. However, we ask that you consider two minor points raised in Referee 2's comments. I expect that these changes can be addressed handily.

We look forward to receiving your revision and thank you for contributing your work to The Journal of Physiology.

REFEREE COMMENTS

Referee #1:

Carefully performed logical series of experiments.

Referee #2:

I am happy with the modifications made.

Very minor possible changes:

Page 13: " Because over such long experimental time courses in some patches substantial inactivation was apparent, currents in the three subsequent sections (P-ATP, ATP, P-ATP) of these experiment were quantitated separately, each normalized to that in the presence of PKA in the respective section." I would replace "analyzed" in place of "quantitated".

Page 18: " Such behaviour was observed neither for WT (Mihályi et al., 2024) nor for Δ F508 CFTR (Fig. 5A), suggesting that activation of the catalytic current component is indeed rate limited by phosphorylation for both WT and the mutant." I would say " Such behaviour was not observed for either WT (Mihályi et al., 2024) or Δ F508 CFTR (Fig. 5A), suggesting that activation of the catalytic current component is indeed rate limited by phosphorylation for both WT and the mutant."

END OF COMMENTS

We have made the requested small revisions.

Dear Dr Csanády,

Re: JP-RP-2026-290798R2 "Altered functional interactions between CFTR disease mutants DeltaF508 and G551D and the protein kinase A catalytic subunit" by Olivér Závoti, Márton A Simon, and László Csanády

We are pleased to tell you that your paper has been accepted for publication in The Journal of Physiology.

Yours sincerely,

Peying Fong
Senior Editor
The Journal of Physiology

IMPORTANT POINTS TO NOTE FOLLOWING ACCEPTANCE OF YOUR PAPER:

- **IMPORTANT NOTICE ABOUT OPEN ACCESS:** To assist authors whose funding agencies mandate immediate public access to published research findings, The Journal of Physiology allows authors to pay an Open Access (OA) fee to have their papers made freely available immediately on publication.

The Corresponding Author will receive an email from Wiley with details on how to register or log in to Wiley Authors where you will be able to place an order.

- You can check if your funder or institution has a Wiley Open Access Account here:
<https://authors.wiley.com/author-resources/Journal-Authors/open-access/author-compliance-tool.html>

- You can help your research get the attention it deserves! Check out Wiley's free Promotion Guide for best-practice recommendations for promoting your work at: www.wileyauthors.com/eeo/guide. You can learn more about Wiley Editing Services which offers professional video, design, and writing services to create shareable video abstracts, infographics, conference posters, lay summaries, and research news stories for your research at: www.wileyauthors.com/eeo/promotion.

- If you would like to receive our 'Research Roundup', a monthly newsletter highlighting the cutting-edge research published in The Physiological Society's family of journals (The Journal of Physiology, Experimental Physiology, Physiological Reports, The Journal of Nutritional Physiology and The Journal of Precision Medicine: Health and Disease), please click this link, fill in your name and email address and select 'Research Roundup':
<https://www.physoc.org/journals-and-media/membernews>

EDITOR COMMENTS

Thank you for incorporating the suggested changes.

Please accept my congratulations. We value your continued contributions to The Journal of Physiology.